

**Transient Dynamics of Terrestrial Carbon Storage: Mathematical foundation and Numeric**
**Examples**
Yiqi Luo[1,2], Zheng Shi[1], Xingjie Lu[3], Jianyang Xia[4], Junyi Liang[1], Jiang Jiang[1], Ying Wang[5],
Matthew J. Smith[6], Lifen Jiang[1], Anders Ahlström[7, 8], Benito Chen[9], Oleksandra Hararuk[10], Alan
Hastings[11], Forrest Hoffman[12], Belinda Medlyn[13], Shuli Niu[14], Martin Rasmussen[15], Katherine
Todd-Brown[16], Ying-Ping Wang[3]
[1]Department of Microbiology and Plant Biology, University of Oklahoma, Norman, Oklahoma,
USA, [2]Center for Earth System Science, Tsinghua University, Beijing, China, [3]CSIRO Oceans
and Atmosphere, Aspendale, Victoria, Australia, [4]School of Ecological and Environmental
Sciences, East China Normal University, Shanghai, China, [5]Department of Mathematics,
University of Oklahoma, Norman, Oklahoma, USA, [6]Computational Science Laboratory,
Microsoft Research, Cambridge, UK, [7]Department of Earth System Science, Stanford
University, Stanford, California, USA, [8]Department of Physical Geography and Ecosystem
Science, Lund University, Lund, Sweden, [9]Department of Mathematics, University of Texas,
Arlington, TX, USA, [10]Pacific Forestry Centre, Canadian Forest Service, Victoria, British
Columbia, Canada, [11]Department of Environmental Science and Policy, University of California,
One Shields Avenue, Davis, CA 95616, USA, [12]Computational Earth Sciences Group, Oak
Ridge National Laboratory, Oak Ridge, TN 37831, USA, [13]Hawkesbury Institute for the
Environment, Western Sydney University, Penrith NSW 2751, Australia, [14]Institute of
Geographic Sciences and Natural Resources Research, Chinese Academy of Sciences, China,
[15]Department of Mathematics, Imperial College, London, UK, [16]Biological Sciences Division,



Pacific Northwest National Laboratory, Richland, Washington, USA,
**Running Title**:  Land carbon storage dynamics
**Correspondence author**: Yiqi Luo
Email: yluo@ou.edu
**Key words** Carbon cycle, carbon sequestration, dynamic disequilibrium, model intercomparison,
terrestrial ecosystems, traceability analysis,
**Type of paper**: Primary Research Article



**Abstract** Terrestrial ecosystems absorb roughly 30% of anthropogenic $CO_2$ emissions since
preindustrial era, but it is unclear whether this carbon (C) sink will endure into the future.
Despite extensive modeling, experimental, and observational studies, what fundamentally
determines transient dynamics of terrestrial C storage under climate change is still not very clear.
Here we develop a new framework for understanding transient dynamics of terrestrial C storage
through mathematical analysis and numerical experiments. Our analysis indicates that the
ultimate force driving ecosystem C storage change is the C storage capacity, which is jointly
determined by ecosystem C input (e.g., net primary production, NPP) and residence time. Since
both C input and residence time vary with time, the C storage capacity is time-dependent and
acts as a moving attractor that actual C storage chases. The rate of change in C storage is
proportional to the C storage potential, the difference between the current storage and the storage
capacity. The C storage capacity represents instantaneous responses of the land C cycle to
external forcing, whereas the C storage potential represents the internal capability of the land C
cycle to influence the C change trajectory in the next time step. The influence happens through
redistribution of net C pool changes in a network of pools with different residence times.

Moreover, this and our other studies have demonstrated that one matrix equation can

exactly replicate simulations of most land C cycle models (i.e., physical emulators). As a result,
simulation outputs of those models can be placed into a three-dimensional (3D) parameter space
to measure their differences. The latter can be decomposed into traceable components to track
the origins of model uncertainty. Moreover, the emulators make data assimilation
computationally feasible so that both C flux- and pool-related datasets can be used to better
constrain model predictions of land C sequestration. We also propose that the C storage potential
be the targeted variable for research, market trading, and government negotiation for C credits.



## 1 Introduction

Terrestrial ecosystems have been estimated to sequester approximately 30% of anthropogenic carbon (C) emission in the past three decades (Canadell et al., 2007). Cumulatively, land ecosystems have sequestered more than 160 Gt C from 1750 to 2015 (Le Quéré et al., 2015). Without land C sequestration, the atmospheric $CO_2$ concentration would have increased by additional 95 parts per million and result in more climate warming (Le Quéré et al., 2015). During one decade from 2005 to 2014, terrestrial ecosystems sequestrated 3±0.8 Gt C per year (Le Quéré et al., 2015), which would cost billion dollars if the equivalent amount of C was sequestrated using C capture and storage techniques (Smith et al., 2016). Thus, terrestrial ecosystems effectively mitigate climate change through natural processes with minimal cost. Whether this terrestrial C sequestration would endure into the future, however, is not clear, making the mitigation of climate change greatly uncertain. To predict future trajectories of C sequestration in the terrestrial ecosystems, it is essential to understand fundamental mechanisms that drive terrestrial C storage dynamics.

To predict future land C sequestration, the modeling community has developed many C cycle models. According to a review by Manzoni and Porporato (2009), approximately 250 biogeochemical models have been published over a time span of 80 years to describe carbon and nitrogen mineralization. The majority of those 250 models follow some mathematical formulations of ordinary differential equations. Moreover, many of those biogeochemical models incorporate more and more processes in an attempt to simulate C cycle processes as realistically as possible (Oleson et al., 2013). As a consequence, terrestrial C cycle models have become increasingly complicated and less tractable. Almost all model intercomparison projects (MIPs), including those involved in the last three IPCC assessments, indicate that C cycle models have



consistently projected widely spread trajectories of land C sinks and also found to fit
observations poorly (Todd-Brown et al., 2013; Luo et al., 2015). The lack of progress in
uncertainty analysis urges us to understand mathematical foundation of those terrestrial C models
so as to diagnose causes of model spreads and improve model predictive skills.

Meanwhile, many countries have made great investments on various observational and

experimental networks (or platforms) in hope to quantify terrestrial C sequestration. For
example, FLUXNET has been established about 20 years ago to quantify net ecosystem
exchange (NEE) between the atmosphere and biosphere (Baldocchi et al., 2001). Orbiting
Carbon Observatory 2 (OCO-2) satellite was launched in 2014 to quantify carbon dioxide
concentrations and distributions in the atmosphere at high spatiotemporal resolution to constrain
land surface C sequestration (Hammerling et al., 2012). Networks of global change experiments
have been designed to uncover processes that regulate ecosystem C sequestration (Rustad et al.,
2001; Luo et al., 2011; Fraser et al., 2013; Borer et al., 2014). Massive data has been generated
from those observational systems and experimental networks. They offer an unprecedented
opportunity for advancing our understanding of ecosystem processes and constraining model
prediction of ecosystem C sequestration. Indeed, many of those networks were initiated with one
goal to improve our predictive capability. Yet the massive data have been rarely integrated into
earth system models to constrain their predictions. It is a grand challenge in our era to develop
innovative approaches to integration of big data into complex models so as to improve prediction
of future ecosystem C sequestration.

From a system perspective, ecosystem C sequestration occurs only when the terrestrial C

cycle is in a transient state, under which C influx into one ecosystem is larger than C efflux from
the ecosystem. Olson (1963) is probably among the first to examine organic matter storage at





forest floors from the system perspective. His analysis approximated steady-state storage of
organic matter as a balance of litter producers and decomposers for different forest types.
However, climate change differentially influences different C cycle processes in ecosystems and
results in transient dynamics of terrestrial C storage (Luo and Weng, 2011). For example, rising
atmospheric $CO_2$ concentration primarily stimulates photosynthetic C uptake while climate
warming likely enhances decomposition. When ecosystem C uptake increases in a unidirectional
trend under elevated $[CO_2]$, terrestrial C cycle is at disequilibrium, leading to net C storage. The
net gained C is first distributed to different pools, each of which has a different turnover rate (or
residence time) before C is eventually released back to the atmosphere via respiration.
Distribution of net C exchange to multiple pools with different residence times is an intrinsic
property of an ecosystem to gradually equalize C efflux with influx (i.e. internal recovery force
toward an attractor). In contrast, climate change that causes changes in C input and
decomposition is considered external forces that create disequilibrium through altering internal C
processes and pool sizes. The transient dynamics of terrestrial C cycle at disequilibrium is
maintained by interactions of internal processes and external forces (Luo and Weng, 2011).
Although the transient dynamics of terrestrial C storage have been conceptually discussed, we
still lack a quantitative formulation to estimate transient C storage dynamics in the terrestrial
ecosystems.

This paper was designed to address a question: what determines transient dynamics of C

storage in terrestrial ecosystems from a system perspective? We first reviewed the major
processes that most models have incorporated to simulate terrestrial C sequestration. The review
helps establish that terrestrial C cycle can be mathematically represented by a matrix equation.
We also described the Terrestrial ECOsystem (TECO) model with its numerical experiments in



support of the mathematical analysis. We then presented results of mathematical analysis on
determinants of the terrestrial C storage, direction and magnitude of C storage at a given time
point, numerical experiments to illustrate climate impacts on terrestrial C storage. We carefully
discussed assumptions of those terrestrial C cycle models as represented by the matrix equation,
the validity of this analysis, and two new concepts introduced in this study, which are the C
storage capacity and C storage potential. We also discussed the potential applications of this
analysis to model uncertainty analysis and data-model integration. Moreover, we proposed that
the C storage potential be a targeted variable for research, trading, and government negotiation
for C credit.

**2 Methods**
**2.1 Mathematical representation of terrestrial C cycle**
This study was conducted mainly with mathematical analysis. We first established the
basis of this analysis, which is that the majority of terrestrial C cycle models can be represented
by a matrix equation.
Hundreds of models have been developed to simulate terrestrial C cycle (Manzoni and
Porporato, 2009). All the models have to simulate processes of photosynthetic C input, C
allocation and transformation, and respiratory C loss. It is well understood that photosynthesis is
a primary pathway of C flow into land ecosystems. Photosynthetic C input is usually simulated
according to carboxylation and electron transport rates (Farquhar et al., 1980). Ecosystem C
influx varies with time and space mainly due to variations in leaf photosynthetic capacity, leaf
area index of canopy, and a suite of environmental factors such as temperature, radiation, and
relative humidity (or other water-related variables) (Potter et al., 1993; Sellers et al., 1996;



Keenan et al., 2012; Walker et al., 2014).
Photosynthetically assimilated C is partly used for plant biomass growth and partly
released back into the atmosphere through plant respiration. Plant biomass in leaves and fine
roots usually lives for several months up to a few years before death, while woody tissues may
persist for hundreds of years in forests. Dead plant materials are transferred to litter pools and
decomposed by microorganisms to be partially released through heterotrophic respiration and
partially stabilized to form soil organic matter (SOM).  SOM can store C in the soil for hundreds
or thousands of years before it is broken down to $CO_2$ through microbial respiration (Luo and
Zhou, 2006). This series of C cycle processes has been represented in most ecosystem models
with multiple pools linked by C transfers among them (Jenkinson et al., 1987; Parton et al., 1987;
1988; 1993), including those embedded in earth system models (Ciais et al., 2013).
The majority of the published 250 terrestrial C cycle models use ordinary differential
equations to describe C transformation processes among multiple plant, litter, and soil pools
(Manzoni and Porporato, 2009). Those ordinary differential equations can be summarized into a
matrix formula (Luo et al., 2003; Luo and Weng, 2011; Luo et al., 2015; 2016; Sierra and Müller
2015) as:
$$X'(t) = Bu(t) - A\xi(t)KX(t) \qquad\qquad (1)$$
where $X'(t)$ is a vector of net C pool changes at time $t$, $X(t)$ is a vector of pool sizes, $B$ is a vector
of partitioning coefficients from C input to each of the pools, $u(t)$ is C input rate, $A$ is a matrix of
transfer coefficients (or microbial C use efficiency) to quantify C movement along the pathways,
$K$ is a diagonal matrix of exit rates (mortality for plant pools and decomposition coefficients of
litter and soil pools) from donor pools and $\xi(t)$ is a diagonal matrix of environmental scalars to
represent responses of C cycle to changes in temperature, moisture, nutrients, litter quality, and



soil texture. In eq. 1, all the off-diagonal $a_{ji}$ values are negative. The equation describes net C
pool change, $X'(t)$, as a result of C input, $u(t)$, distributed to different plant pools via
partitioning coefficients, $B$, minus C loss through C transformation matrix, $A\xi(t)K$, among
individual pools, $X(t)$. Elements in vector $B$, matrices $A$ and $K$ could vary with many factors,
such as vegetation types, soil textual, microbial attributes, and litter chemistry. For example,
vegetation succession may influence elements in vector $B$, matrices $A$ and $K$ in addition to C
input, $u(t)$, and forcing that affects C dynamics through environmental scalars, $\xi(t)$.

After synthesis of all the possible soil C cycle models based on six principles (mass

balance, substrate dependence of decomposition, heterogeneity of decay rates, internal
transformations of organic matter, environmental variability effects, and substrate interactions),
Sierra and Müller (2015) concluded that this form of matrix equation such as eq. 1 represents the
majority of terrestrial C cycle models. Similarly, Manzoni and Porporato (2009) concluded their
review of 250 models that the majority of them use ordinary differential equations, which can be
summarized by eq. 1, to describe land C cycle. Our mathematical analysis in this study used
matrix operations of eq. 1 to reveal determinants of transient dynamics of terrestrial C cycle,
including direction and rate of C storage changes, in response to climate change. We examined
assumptions underlying this equation and the validity of our analysis in the Discussion section.

**2.2 Model and its numerical experiments**

We conducted numerical experiments to support the mathematical analysis and thus help

understand the characteristics of terrestrial C storage dynamics using the Terrestrial ECOsystem
(TECO) model. TECO has five major components: canopy photosynthesis, soil water dynamics,
plant growth, litter and soil carbon decomposition and transformation, and nitrogen dynamics as





described in detail by Weng and Luo (2008) and Shi et al. (2016). Canopy photosynthesis is
referred from a two-leaf (sunlit and shaded) model developed by Wang and Leuning (1998). This
submodel simulates canopy conductance, photosynthesis, and partitioning of available energy.
The model combines the leaf photosynthesis model developed by Farquhar et al. (1980) and a
stomatal conductance model (Harley et al., 1992). In the soil water dynamic submodel, soil is
divided into 10 layers. The surface layer is 10 cm deep and the other 9 layers are 20 cm deep.
Soil water content (SWC) in each layer results from the mass balance between water influx and
efflux. The plant growth submodel simulates C allocation and phenology. Allocation of C among
three plant pools, which are leaf, fine root, and wood, depends on their growth rates (Fig. 1a).
Phenology dynamics is related to leaf onset, which is triggered by growing degree days, and leaf
senescence, which is determined by temperature and soil moisture. The C transformation
submodel estimates carbon transfer from plants to two litter pools and three soil pools (Fig. 1a).
The nitrogen (N) submodel is fully coupled with C processes with one additional mineral N pool.
Nitrogen is absorbed by plants from mineral soil and then partitioned among leaf, woody tissues
and fine roots. Nitrogen in plant detritus is transferred among different ecosystem pools (i.e.
litter, coarse wood debris, fast, slow and passive SOM) (Shi et al., 2016). The model is driven by
climate data, which included air and soil temperature, vapor-pressure deficit, relative humidity,
incident photosynthetically active radiation, and precipitation at hourly steps.

We first calibrated TECO with eddy flux data collected at Harvard Forest from 2006-

2009. The calibrated model was spun up to the equilibrium state in pre-industrial environmental
conditions by recycling a 10-year climate forcing (1850-1859). Then the model was used to
simulate C dynamics from year 1850 to 2100 with the historical forcing scenario for 1850-2005
and RCP8.5 scenario for 2006-2100 as in the Community Land Model 4.5 (Oleson et al., 2013)





in the grid cell where Harvard Forest is located.

To support the mathematical analysis using eq. 1, we first verified that eq. 1 can exactly

represent TECO model simulations. We first identified those variables in each of the C balance
equations in the TECO model that are corresponding to elements in matrices $A$, $\xi(t)$, and $K$, and
vectors $X(t)$, and $B$ together with variable $u(t)$ in eq. 1. Then we ran the TECO model to
generate outputs of all those variables at each time step, which were consequently organized into
matrices $A$, $\xi(t)$, and $K$, and vectors $X(t)$ and $B$, and variable $u(t)$. Those matrices, vectors, and
variable were entered to matrix calculation to compute $X'(t)$ using eq. 1. The sum of elements in
calculated $X'(t)$ is a 100% match with simulated net ecosystem production (NEP) with the
TECO model (Fig. 1b).

Once eq. 1 was verified to exactly replicate TECO simulations, we use TECO to generate

numerical experiments to support the mathematical analysis on the transient dynamics of
terrestrial C storage. To analyze the seasonal patterns of C storage dynamics, we averaged 10
series of three-year seasonal dynamics from 1851-1880. Then we used a 7-day moving window
to further smooth the data.

**3. Results**

**3.1 Determinants of C storage dynamics**
The transient dynamics of terrestrial carbon storage are determined by two components: the C
storage capacity and the C storage potential. The two components of C storage dynamics can be
mathematically derived from multiplying both sides of eq. 1 by $(A\xi(t)K)^{-1}$ as:

$X(t) = (A\xi(t)K)^{-1}Bu(t) - (A\xi(t)K)^{-1}X'(t)$                (2)



The first term on the left side of eq. 2 is the C storage capacity and the second term is the C
storage potential. Fig. 2a shows time courses of C storage and its capacity over one year for the
leaf pool of Harvard Forest.
In eq. 2, we name the term $(A\xi(t)K)^{-1}$ the chasing time, $\tau_{ch}(t)$, as:
$$\tau_{ch}(t) = (A\xi(t)K)^{-1} \qquad (3)$$
$\tau_{ch}(t)$ is a matrix of C residence times through the network of individual pools each with
different capacities as measured by their residence times and fractions of received C connected
by pathways of C transfer. Analogous to the fundamental matrix measuring life expectancies in
demographic models (Caswell, 2000), the matrix, $\tau_{ch}(t)$, here measures expected residence time
of a C atom in pool $i$ when it has entered from pool $j$. We call this matrix the fundamental matrix
of chasing times to represent the time scale at which the net C pool change, $X'(t)$, is
redistributed in the network. Meanwhile, the residence time of individual pools in network can
be estimated by multiplying the fundamental matrix of chasing times, $(A\xi(t)K)^{-1}$, by a vector
of partitioning coefficients, $B$ as:
$$\tau_E(t) = (A\xi(t)K)^{-1}B \qquad (4)$$
Ecosystem residence time is the sum of the residence time of all individual pools in network,
Thus, the C storage capacity can be defined by:
$$X_c(t) = (A\xi(t)K)^{-1}Bu(t) \qquad (5a)$$
Or it can be estimated from input C, $u(t)$, and residence time, $\tau_E(t)$, as:
$$X_C(t) = \tau_E(t)u(t) \qquad (5b)$$
As C input (e.g., Gross or Net Primary Productions, GPP or NPP) and residence times vary with
time, the C storage capacity varies with time. It represents instantaneous responses of the
terrestrial C cycle to the external forcing. The modeled C storage capacity in the leaf pool (Fig.





2a), for example, increases in spring, reaches the peak at summer, declines in autumn, and
becomes minimal in winter largely due to strong seasonal changes in C input (Fig. 2b). Note that
either GPP or NPP can be used as C input for analysis of transient C dynamics. Estimated
residence times, however, are smaller with GPP as C input than those with NPP as input. In this
paper, we mostly used NPP as C input as that fraction of C is distributed among pools.
The C storage potential at time $t$, $X_p(t)$, can be mathematically described as:
$$X_p(t) = (A\xi(t)K)^{-1}X'(t) \qquad (6a)$$
Or it can be estimated from net C pool change, $X'(t)$, and chasing time, $\tau_{ch}(t)$ as:
$$X_p(t) = \tau_{ch}(t)X'(t) \qquad (6b)$$
Eqs. 6a and 6b suggest that the C storage potential represents re-distribution of net C pool
change, $X'(t)$, of individual pools through a network of pools with different residence times as
connected by C transfers from one pool to the others through all the pathways. As time evolves,
the net C pool change, $X'(t)$, is redistributed again and again through the network of pools. The
network of redistribution of next C pool change, thus, represents the potential of an ecosystem to
store additional C when it is positive and lose C when it is negative. The C storage potential can
also be estimated from the difference between the C storage capacity and the C storage itself at
time $t$ as:
$$X_p(t) = X_c(t) - X(t) \qquad (6c)$$
The C storage potential in the leaf pool, for example, is about zero in winter and early spring
when the C storage capacity is very close to the storage itself (Fig. 2a). The C storage potential is
positive when the capacity is larger than the storage itself from late spring to summer and early
fall. As the storage capacity decreases to the point when the storage equals the capacity on the
265[th] day of year (DOY), the C storage potential is zero. After that day, the C storage potential



becomes negative.

Dynamics of ecosystem C storage, $X(t)$, can be characterized by three parameters: C

influx, $u(t)$, residence times, $\tau_E(t)$, and the C storage potential $X_p(t)$ as:

$$X(t) = \tau_E(t)u(t) - X_p(t) \qquad (7)$$

Eq. 7 represents a three-dimensional (3D) parameter space within which model simulation
outputs can be placed to measure how and how much they diverge.

Note that sums of elements in vectors $X(t)$, $X_c(t)$, $X_p(t)$, $X'(t)$, and $\tau_E(t)$ are

corresponding, respectively, to the whole ecosystem C stock, ecosystem C storage capacity,
ecosystem C storage potential, net ecosystem production (NEP), and ecosystem residence time.
In this paper, we do not use a separate set of symbols to represent those sums rather than express
them wherever necessary.

**3.2 Direction and rate of C storage change at a given time**
Like studying any moving object, quantifying dynamics of land C storage needs to determine
both the direction and the rate of its change at a given time. To determine the direction and rate
of C storage change, we re-arranged eq. 2 to be:

$$\tau_{ch}X'(t) = X_c(t) - X(t) = X_p(t) \qquad (8a)$$

or re-arranging eq. 6a leads to:

$$X'(t) = A\xi(t)KX_p(t) \qquad (8b)$$

As all the elements in $\tau_{ch}$ are positive, the sign of $X'(t)$ is the same as for $X_p(t)$. That means
$X'(t)$ increases when $X_c(t) > X(t)$, does not change when $X_c(t) = X(t)$, and decreases when
$X_c(t) < X(t)$ at the ecosystem scale. Thus, the C storage capacity, $X_c(t)$, is an attractor and





hence determines the direction toward which the C storage, $X(t)$, chases at any given time point.
The rate of C storage change, $X'(t)$, is proportional to $X_p(t)$ and also regulated by $\tau_{ch}$.
When we study C cycle dynamics, we are not only interested in understanding dynamics
of a whole ecosystem but also individual pools. Eq. 8a can be used to derive equations to
describe C storage change for an i$^{\text{th}}$ pool as:
$\qquad \sum_{j=1}^{n} f_{ij}\, \tau_i\, x'_j(t) = \sum_{j=1}^{n} f_{ij}\, \tau_i b_j u(t) - x_i(t) = x_{p,i}(t)$ $\qquad$ (9a)
where $n$ is the number of pools in a C cycle model, $f_{ij}$ is a fraction of C transferred from pool j
to i through all the pathways, $\tau_i$ measure residence times of individual pools in isolation, $x'_j$ is
the net C change in the j$^{\text{th}}$ pool, $b_j$ is a partitioning coefficient of C input to the j$^{\text{th}}$ pool, $x_i(t)$ is
the C storage in the i$^{\text{th}}$ pool, and $x_{p,i}(t)$ is the C storage potential in the i$^{\text{th}}$ pool. Eq. 9a means
that the C storage potential of each pool at time $t$, $x_{p,i}(t)$, is the sum of all the individual net C
pool change, $x'_j$, multiplied by corresponding residence time spent in pool $i$ coming from pool $j$.
Through re-arrangement, eq. 9a can be solved for each individual pool net C change as a
function of C storage potential of all the pools as:
$\qquad x'_i(t) = \dfrac{x_{c,i,u}(t) - x_{c,i,p}(t) - x_i(t)}{f_{ii}\tau_i}$ $\qquad$ (9b)
where $x_{c,i,u}(t) = \sum_{j=1}^{n} f_{ij}\, \tau_i b_j u(t)$ for the maximal amount of C that can transfer from C input
to the i$^{\text{th}}$ pool. $x_{c,i,p}(t) = \sum_{j=1,j\neq i}^{n} f_{ij}\, \tau_i x'_j(t)$ for the maximal amount of C that can transfer from
all the other pools to the i$^{\text{th}}$ pool. $f_{ii} = 1$ for all the pools if there is no feedback of C among soil
pools. $f_{ii} < 1$ when there are feedbacks of C among soil pools.
As plant pools get C only from photosynthetic C input, $u(t)$, but not from other pools,
the direction and rate of C storage change in the i$^{\text{th}}$ plant pool is determined by:



$$\begin{cases} x'_i(t) = \dfrac{x_{c,i}(t) - x_i(t)}{\tau_i} = \dfrac{X_{p,i}(t)}{\tau_i} \\ x_{c,i}(t) = b_i u(t) \tau_i \end{cases} \quad \text{for i = 1, 2, 3} \quad (10)$$

The C storage capacity of plant pools equals the product of plant C input, $u(t)$ (i.e., net primary
production, NPP), partitioning coefficient, $b_i$, and residence time, $\tau_i$, of its own pool (Fig. 2b-d).
Thus, the C storage capacities of the leaf, root, and wood pools are high in summer and low in
winter. Plant C storage, $x_i(t)$, still chases the storage capacity, $x_{c,i}(t)$, of its own pool at a rate
that is proportional to $X_{p,i}(t)$. For the leaf pool, the C storage, $x_1(t)$, increases when $x_{c,1}(t) >$
$x_1(t)$ (or $x_{p,1}(t)>0$) from late spring until early fall on the 265th day of year (DOY) and then
decreases when $x_{c,1}(t) < x_1(t)$ (or $x_{p,1}(t)<0$) from DOY of 265 until 326 during fall (Fig. 2a).

However, the direction of C storage change in litter and soil pools are no longer solely

determined by the storage capacity, $x_{c,i}(t)$, of their own pools or at a rate that is proportional to
$X_{p,i}(t)$. The C storage capacity of one litter or soil pool has two components. One component,
$x_{c,i,u}(t)$ is set by the amount of plant C input, $u(t)$, going through all the possible pathways,
$f_{ij}b_j$, multiplied by residence time, $\tau_i$, of its own pool.  The second component measures the C
exchange of one litter or soil pool with other pools according to net C pool change, $x'_j(t)$,
through pathways, $f_{ij}, j \neq i$, weighed by residence time, $\tau_i$, of its own pool. For example, C
input to the litter pool is a combination of C transfer from C input through the leaf, root, and
wood pools (Fig. 3c, 3d, and 3e) and C transfer due to the net C pool changes in the leaf, root,
and wood pools (Fig. 3f, 3g, and 3h). Thus the first capacity component of the litter pool to store
C is the sum of three products of NPP, C partitioning coefficient, and network residence time,
respectively, through the leaf, root, and wood pools (Fig. 3c, 3d, and 3e). The second capacity
component is the sum of other three products of C transfer coefficient along all the possible
pathways, network residence time, and net C pool changes, respectively, in the leaf, root, and



wood pools (Fig. 3f, 3g, and 3h). Thus, C storage in the $i^{th}$ pool, $x_i(t)$, chases an attractor,
$(\sum_{j=1}^{n} f_{ij} b_j u(t) - \sum_{j=1,j \neq i}^{n} f_{ij} \tau_i x'_j(t))\tau_i$, for litter and soil pools (Fig. 4).

In summary, due to the network of C transfer, C storage in litter and soil pools does not

chase the C storage capacities of their own pools in a multiple C pool model (Fig. 4). The
capacities for individual litter and soil pools measure the amounts of C that is transferred from
photosynthetic C input through plant pools to be stored in those pools. However, those litter and
soil pools also exchange C with other pools according to transfer coefficients along pathways of
C movement multiplying net C pool change in those pools. Integration of the C input and C
exchanges together still set as a moving attractor toward which individual pool C storage
approaches (Fig. 4).

**365 3.3 C storage dynamics under climate change**

In response to a climate change scenario that combines historical change and simulated RCP8.5
in the TECO experiment, the modeled ecosystem C storage capacity (the sum of all elements in
vector $X_c(t)$) at Harvard Forest increases from 27 kg C m$^{-2}$ in 1850 to approximately 38 kg C m$^{-2}$
in 2100 with strong interannual variability (Fig. 5a). The increasing capacity results from a
combination of a nearly 44% increase in NPP with a ~2% decrease in ecosystem residence times
(the sum of all elements in vector $\tau_E(t)$) during that period (Fig. 5b). The strong interannual
variability in the modeled capacity is attributable to the variability in NPP and residence times,
both of which directly respond to instantaneous variations in environmental factors. In
comparison, the ecosystem C storage (the sum of all elements in vector $X(t)$) itself gradually
increases, lagging behind the capacity, with much dampened interannual variability (Fig. 5a).
The dampened interannual variability is due to smoothing effects of pools with various residence



times. In response to climate change scenario RCP8.5, the ecosystem C storage potential (the
sum of all elements in vector $X_p(t)$) in the Harvard Forest ecosystem increases from zero at
1980 to 3.5 kg C m$^{-2}$ in 2100 with strong fluctuation over years (Fig. 5a).  Over seasons, the
potential is high during the summer and low in winter, similarly with the seasonal cycle of the C
storage capacity.

Since chasing time, $\tau_{ch}$, is a matrix and net C pool change, $X'(t)$, is a vector, eq. 6a or 6b

(i.e., the C storage potential) can not be analytically separated into the chasing time and net C
pool change as can the capacity into C input and residence time in eq. 5a or 5b for traceability
analysis. The relationships among the three quantities can be explored by regression analysis.
The ecosystem C storage potential fluctuates in a similar phase with NEP from 1850 to 2100
(Fig. 5c). Consequently, the C storage potential is well correlated with NEP at the whole
ecosystem scale (Fig. 5d).  The slope of the regression line is a statistical representation of
ecosystem chasing time. In this study, we find that r$^2$ of the relationship between the storage
potential and NEP is 0.79. The regression slope is 28.1 years in comparison with the ecosystem
residence time of approximately 22 years (Fig. 5b).

The capacity and storage itself of individual pools display similar long-term trends and

interannual variability to those for the total ecosystem C storage dynamics (Fig. 6). Noticeably,
the deviation of the C storage from the capacity, which is the C storage potential, is much larger
for pools with long residence times than those with short residence times. For individual pools,
the potential is nearly zero for those fast turnover pools and becomes very large for those pools
with long residence time (Fig. 6).

For individual plant pools, eq. 10 describes the dependence of the C storage potential,

$x_{p,i}(t)$, on the pool-specific residence time, $\tau_i$, $i$ = 1, 2, and 3, and net C pool change of their



own pools, $x'_i(t)$, $i$ = 1, 2, and 3. Thus, one value of $x_{p,i}(t)$ is exactly corresponding to one
value of $x'_i(t)$ at slope of $\tau_i$, leading to the correlation coefficient in Fig. 7 being 1.00 for leaf,
root, and wood pools. For a litter or soil pool, however, the C storage potential is not solely
dependent on the residence time and net C pool change of its own pool but influenced by several
other pools. Thus, the potential of one litter or soil pool is correlated with net C pool changes of
several pools with different regression slopes (Fig. 7).

**4 Discussion**
4.1 Assumptions of the C cycle models and validity of this analysis

This analysis is built upon eq. 1, which represents the majority of terrestrial C cycle

models developed in the past decades (Manzoni and Porporato, 2009; Sierra and Müller, 2015).
Those models have several assumptions, which may influence the validity of this analysis. First,
those models assume that donor pools control C transfers among pools and decomposition
follows 1st-order decay functions (Assumption 1). This assumption is built upon observations
from litter and SOC decomposition. Analysis of data from nearly 300 studies of litter
decomposition (Zhang et al., 2008), about 500 studies of soil incubation (Xu et al., 2016), more
than 100 studies of forest succession (Yang et al., 2011), and restoration (Matamala et al., 2008)
almost all suggest that the 1st-order decay function captures macroscopic patterns of land C
dynamics. Even so, its biological, chemical and physical underpinnings need more study (Luo et
al., 2016). This assumption has recently been challenged by a notion that microbes are actively
involved in decomposition processes. To describe the active roles of microbes in organic C
decomposition, a suite of nonlinear microbial models has been proposed using Michaelis-Menten
or reverse Michaelis-Menten equations (Allison et al., 2010; Wieder et al., 2013). Those




nonlinear models exhibit unique behaviors of modeled systems, such as damped oscillatory
responses of soil C dynamics to small perturbations and insensitivity of the equilibrium pool
sizes of litter or soil carbon to inputs (Li et al., 2014; Wang et al., 2014; 2016). Oscillations have
been documented for single enzymes at timescales between $10^{-4}$ to 10 seconds (English et al.,
2006; Goldbeter, 2013; Xie, 2013). Over longer timescales with mixtures of large diversity of
enzyme-substrate complexes in soil, oscillations may be likely averaged out so that the 1st order
decay functions may well approximate these average dynamics of organic matter decomposition
(Sierra and Müller, 2015).

Second, those models all assume that multiple pools can adequately approximate

transformation, decomposition, and stabilization of SOC in the real world (Assumption 2). The
classic SOC model, CENTURY, uses three conceptual pools, active, slow, and passive SOC, to
represent SOC dynamics (Parton et al., 1987). Several models define pools that are
corresponding to measurable SOC fractions to match experimental observation with modeling
analysis (Smith et al., 2002; Stewart et al., 2008). Carbon transformation in soil over time has
also been described by a partial differential function of SOM quality (Bosatta and Ågren, 1991;
Ågren and Bosatta, 1996). The latter quality model describes the external inputs of C with
certain quality, C loss due to decomposition, and the internal transformations of the quality of
soil organic matter. It has been shown that multi-pool models can approximate the partial
differential function or continuous quality model as the number of pools increases (Bolker et al.,
1998; Sierra and Müller, 2015).

Assumption 3 is on partitioning coefficients of C input (i.e., elements in vector $B$) and C

transformation among plant, litter, and soil pools (i.e., elements in the matrix, $A\xi(t)K$). Some of
the terrestrial C cycle models assume that elements in vector $B$, and matrices $A$ and $K$ are



constants. All the factors or processes that vary with time are represented in the diagonal matrix
$\xi(t)$. In the real world, C transformation are influenced by environmental variables (e.g.,
temperature, moisture, oxygen, N, phosphorus, and acidity varying with soil profile, space, and
time), litter quality (e.g., lignin, cellulose, N, or their relative content), organomineral properties
of SOC (e.g., complex chemical compounds, aggregation, physiochemical binding and
protection, reactions with inorganic, reactive surfaces, and sorption), and microbial attributes
(e.g., community structure, functionality, priming, acclimation, and other physiological
adjustments) (Luo et al., 2016). It is not practical to incorporate all of those factors and processes
into one model. Only a subset of them is explicitly expressed while the majority is implicitly
embedded in the C cycle models.  Empirical studies have suggested that temperature, moisture,
litter quality, and soil texture are primary factors that control C transformation processes of
decomposition and stabilization (Burke et al., 1989; Adair et al., 2008; Zhang et al., 2008; Xu et
al., 2012; Wang et al., 2013). Nitrogen influences C cycle processes mainly through changes in
photosynthetic C input, C partitioning, and decomposition. It is yet to identify how other major
factors and processes, such as microbial activities and organomineral protection, regulate C
transformation.

Assumption 4 is that terrestrial C cycle models use different response functions (i.e.,

different $\xi(t)$ in eq. 1) to represent C cycle responses to external variables. As temperature
modifies almost all processes in the C cycle, different formulations, including exponential,
Arrhenius, and optimal response functions, have been used to describe C cycle responses to
temperature changes in different models (Lloyd and Taylor, 1994; Jones et al., 2005; Sierra and
Müller, 2015). Different response functions are used to connect C cycle processes with moisture,
nutrient availability, soil clay content, litter quality, and other factors. Different formulations of



response functions may result in substantially different model projections (Exbrayat et al., 2013)
but unlikely change basic dynamics of the model behaviors.
Assumption 5 is that disturbance events are represented in models in different ways
(Grosse et al., 2011; West et al., 2011; Goetz et al., 2012; Hicke et al., 2012). Fire, extreme
drought, insect outbreaks, land management, and land cover and land use change influence
terrestrial C dynamics via 1) altering rate processes, for example, gross primary productivity
(GPP), growth, tree mortality, or heterotrophic respiration; 2) modifying microclimatic
environments; 3) transferring C from one pool to another (e.g., from live to dead pools during
storms or release to the atmosphere with fire) (Kloster et al., 2010; Thonicke et al., 2010; Luo
and Weng, 2011; Prentice et al., 2011; Weng et al., 2012). Many disturbance events are
incorporated into terrestrial C cycle models without changing the basic formulation (i.e., eq. 1)
(Weng et al., 2012).
The sixth assumption that those models make is that the lateral C fluxes through erosion
or local C drainage is negligible so that eq (1) can approximate terrestrial C cycle over space. If
soil erosion is substantial enough to be modeled with horizontal movement of C, a third
dimension should be added in addition to two-dimensional transfers in classic models.
Our analysis on transient dynamics of terrestrial C cycle is valid unless some of the
assumptions are violated. Assumption 1 on the 1$^{st}$-order decay function of decomposition
appears to be supported by thousands of datasets. It is a burden on microbiologists to identify
empirical evidence to support the nonlinear microbial models. Assumption 2 may not affect the
validity of our analysis no matter how C pools are divided in the ecosystems. Our analysis in this
study is applicable no matter whether elements are time-varying or constant in vector $B$ and
matrices $A$ and $K$ as in assumption 3. Neither assumption 4 nor 5 would affect the analysis in this



study. The environmental scalar, $\xi(t)$, as related to assumption 4 can be any forms in the derived
equations (e.g., eq. 2). Disturbances of fire, land use, and extreme drought change rate processes
but do not alter the basic formulation of eq. 1. If soil erosion and lateral transportation of C
become a major research objective, Eq. (1) can no longer be analyzed to understand the
mathematical foundation underlying transient dynamics of terrestrial C cycle.

**4.2 Carbon storage capacity**

One of the two components this analysis introduces to understand transient dynamics of

terrestrial C storage is the C storage capacity (Eq. 2). Olson (1963) is probably among the first
who systematically analyzed C storage dynamics at forest floor as functions of litter production
and decomposition. He collected data of annual litter production and approximately steady-state
organic C storage at forest floor, from which decomposition rates were estimated for a variety of
ecosystems from Ghana in the tropics to alpine forests in California. Using the relationships
among litter production, decomposition, and C storage, Olson (1963) explored several issues,
such as decay without input, accumulation with continuous or discrete annual litter fall, and
adjustments in production and decay parameters during forest succession. His analysis
approximated the steady-state C storage as the C input times the inverse of decomposition (i.e.,
residence time). The steady-state C storage is also considered the maximal amount of C that a
forest can store.

This study is not only built upon Olson's analysis but also expands it at least in two

aspects. First, we similarly define the C storage capacity (i.e., eqs. 5a and 5b). Those equations
can be applied to a whole ecosystem with multiple C pools while Olson's analysis is for one C
pool. Second, Olson (1963) treated the C input and decomposition rate as yearly constants at a





given location even though they varied with locations. This study considers both C input and rate
of decomposition being time dependent. A dynamical system with its input and parameters being
time dependent mathematically becomes a nonautonomous system (Kloeden and Rasmussen,
2011). As terrestrial C cycle under climate change is transient, we need to treat it a
nonautonomous system to better understand the properties of transient dynamics. Olson (1963)
approximated the non-autonomous system at the yearly time scale without climate change so as
to effectively understand properties of the steady-state C storage at the forest floor. In
comparison, eqs. 5a and b are not only more general but also essential for understanding
transient dynamics of the terrestrial C cycle in response to climate change.

Under the transient dynamics, the C storage capacity as defined by eqs 5a and b still sets

the maximal amount of C that one ecosystem can store at time $t$. This capacity represents
instantaneous responses of ecosystem C cycle to external forcing via changes in both C input and
residence time, and thus varies within one day, over seasons of a year, and interannually over
longer time scales as forcings vary. The variation of the C storage capacity can result from cyclic
environmental changes (e.g., dial and seasonal changes), directional climate change (e.g., rising
atmospheric $CO_2$, nitrogen deposition, altered precipitation, and warming), disturbance events,
disturbance regime shifts, and changing vegetation dynamics (Luo and Weng, 2011). As the
capacity sets the maximal amount of C storage (Fig. 2a), it is a moving attractor toward which
the current C storage chases.  When the capacity is larger than the C storage itself, C storage
increases. Otherwise, the C storage decreases.

**4.3 Carbon storage potential**

The C storage potential represents the internal capability to equilibrate the current C





storage with the capacity. Bogeochemically, the C storage potential represents re-distribution of
net C pool change, $X'(t)$, of individual pools through a network of pools with different residence
times as connected by C transfers from one pool to the others through all the pathways. The
potential is conceptually equivalent to the magnitude of disequilibrium as discussed by Luo and
Weng (2011).

The C storage potential measures the amount of additional C that one ecosystem can

store. Thus it can be used as a targeted quantity for C cycle research, C trading, and C credit in
government negotiation. In many fields of research, there are clearly targeted quantities on which
research would be focused.  For example, crop science primarily focuses on crop yield although
environmental consequences of increasing crop yield have to be quantified.  Gross domestic
product (GDP) is the targeted indicator that a country manages their economy. Although C cycle
has become a major research topic, has markets for trading, and is managed by governments, no
consensus has been established on the targeted quantity that our study should focus on.

Extensive studies have been done to quantify terrestrial C sequestration. The most

commonly estimated quantities for C sequestration include net ecosystem exchange (NEE), C
stocks in ecosystems (i.e., plant biomass and SOC) and their changes (Baldocchi et al., 2001; Pan
et al., 2013). This study, for the first time, offers the theoretical basis to estimate the terrestrial C
storage potential in at least two approaches: (1) the product of chasing time and net C pool
change with eqs. 6a and 6b; and (2) the difference between the C storage capacity and the C
storage itself with eqs. 6c. Since the time-varying C storage capacity is fully defined by
residence time and C input at any given time, C storage potential can be estimated from three
quantities: C input, residence time, and C storage.

To effectively quantify the C storage potential in terrestrial ecosystems, we need various





data sets from experimental and observatory studies to be first assimilated into models. For
example, data from Harvard Forest were first used to constrain the TECO model. The
constrained model was used to explore changes in ecosystem C storage in response to climate
change scenario, RCP8.5. That scenario primarily stimulated NPP, which increased from 1.06 to
1.8 kg C m$^{-2}$ yr$^{-1}$ in the Harvard Forest (Fig. 5b). Although climate warming decreased residence
time in the Harvard Forest, the substantial increases in NPP resulted in increases in the C storage
potential over time.

**4.4 Novel approaches to model evaluation and improvement**

Our analysis of transient C cycle dynamics offers new approaches to understand,

evaluate, diagnose, and improve land C cycle models.  We have demonstrated that many global
land C cycle models can be exactly represented by the matrix equation (Eqs. 1 and 2) (i.e.,
physical emulators). As a consequence, outputs of all those models can be placed into a three
dimensional (3D) space (Eq. 7) to measure their differences. In addition, components of land C
cycle models are simulated in a mutually independent fashion so that modeled C storage can be
decomposed into traceable components for traceability analysis. Moreover, the physical
emulators computationally enable data assimilation to constrain complex models.

*Physical Emulators of land C cycle models* We have developed matrix representations

(i.e., physical emulators) of CABLE, LPJ-GUESS, CLM3.5, CLM 4.0, CLM4.5, BEPS, and
TECO (Xia et al., 2013; Hararuk et al., 2014; Ahlström et al., 2015; Chen et al., 2015). The
emulators can exactly replicate simulations of C pools and fluxes with their original models
when driven by a limited set of inputs from the full model (GPP, soil temperature, and soil
moisture) (Fig. 1b and 1c). The emulators make complex models analytically clear and,



therefore, give us a way to understand the effects of forcing, model structures, and parameters on
modeled ecosystem processes. They greatly simplify the task of understanding the dynamics of
submodels and interactions between them. The emulators allow us analyze model results in the
3D parameter space and the traceability framework.

*Parameter space of C cycle dynamics* Eq. 7 indicates that transient dynamics of modeled

C storage are determined by three parameters: C input, residence time, and C storage potential.
The 3D parameter space offers one novel approach to uncertainty analysis of global C cycle
models. As global land models incorporate more and more processes to simulate C cycle
responses to global change, it becomes very difficult to understand or evaluate complex model
behaviors. As such, differences in model projections cannot be easily diagnosed and attributed to
their sources (Chatfield, 1995; Friedlingstein et al., 2006; Luo et al., 2009). Eq. 7 can help
diagnose and evaluate complex models by placing all modeling results within one common
parameter space in spite of the fact that individual global models may have tens or hundreds of
parameters to represent C cycle processes as affected by many abiotic and biotic factors (Luo et
al., 2016). The 3D space can be used to measure how and how much the models diverge.

*Traceability analysis* The two terms on the right side of eq. 2 can be decomposed into

traceable components (Xia et al., 2013) so as to identify sources of uncertainty in C cycle model
projections. Model intercomparison projects (MIPs) all illustrate great spreads in projected land
C sink dynamics across models (Todd-Brown et al., 2013; Tian et al., 2015).  It has been
extremely challenging to attribute the uncertainty to sources. Placing simulation results of a
variety of C cycle models within one common parameter space can measure how much the
model differences are in a common metrics (Eq. 7). The measured differences can be further
attributed to sources in model structure, parameter, and forcing fields with traceability analysis



(Xia et al., 2013; Rafique et al., 2014; Ahlström et al., 2015; Chen et al., 2015). The traceability
analysis also can be used to evaluate effectiveness of newly incorporated modules into existing
models, such as adding the N module on simulated C dynamics (Xia et al., 2013) and locate the
origin of model ensemble uncertainties to external forcing vs. model structures and parameters
(Ahlström et al., 2015).

*Constrained estimates of terrestrial C sequestration* Traditionally, global land C sink is

indirectly estimated from airborne fraction of C emission and ocean uptake. Although many
global land models have been developed to estimate land C sequestration, a variety of MIPs
indicate that model predictions widely vary among them and do not fit observations well
(Schwalm et al., 2010; Luo et al., 2015; Tian et al., 2015). Moreover, the prevailing practices in
the modeling community, unfortunately, may not lead to significant enhancements in our
confidence on model predictions. For example, incorporating an increasing number of processes
that influence the C cycle may represent the real-world phenomena more realistically but makes
the models more complex and less tractable. MIPs have effectively revealed the extent of the
differences between model predictions (Schwalm et al., 2010; Keenan et al., 2012; De Kauwe et
al., 2013) but provide limited insights into sources of model differences (but see Medlyn et al.
(2015). The physical emulators make data assimilation computationally feasible for global C
cycle models Hararuk *et al*. (2014; 2015) and thus offer the possibility to generate independent
yet constrained estimates of global land C sequestration to be compared with the indirect
estimate. With the emulators, we can assimilate most of the C flux- and pool-related datasets into
those models to better constrain global land C sink dynamics.

**Concluding remarks**





In this study we theoretically explored the transient dynamics of terrestrial C storage. Our
analysis indicates that transient C storage dynamics can be partitioned into two components: the
C storage capacity and the C storage potential. The capacity, which is the product of C input and
residence time, represents their instantaneous responses to a state of external forcing at a given
time. Thus, the C storage capacity quantifies the maximum amount of C that an ecosystem can
store at the given environmental condition at a point of time. Thus it varies diurnally, seasonally,
and interannually as environmental condition changes.

The C storage potential is the difference between the capacity and the current C storage

and thus measures the magnitude of disequilibrium in the terrestrial C cycle (Luo and Weng,
2011). The storage potential represents the internal capability (or recovery force) of the
terrestrial C cycle to influence the change in C storage in the next time step through
redistribution of net C pool changes in a network of multiple pools with different residence
times. The redistribution drives the current C storage towards the capacity and thus equilibrates
C efflux with influx. We propose that the storage potential should be the targeted quantity for
research, market trading, and government management for C credits.

The two components of land C storage dynamics represent interactions of external forces

(via changes in the capacity) and internal capability of the land C cycle (via changes in the C
storage potential) to generate complex phenomena of C cycle dynamics, such as fluctuations,
directional changes, and tipping points, in the terrestrial ecosystems. From a system perspective,
these complex phenomena are mostly caused by multiple environmental forcing variables
interacting with relatively simple internal processes over different temporal and spatial scales.
Note that while those internal processes can be mathematically represented with a relatively
simple formula, their ecological and biological underpinnings can be very complex.



The theoretical framework developed in this study has the potential to revolutionize
model evaluation. Our analysis indicates that the matrix equation as in eq. 1 or 2 can adequately
emulate most of the land C cycle models. Indeed, we have developed physical emulators of
several global land C cycle models. In addition, predictions of C dynamics with complex land
models can be placed in a 3D parameter space as a common metric to measure how much model
predictions are different. The latter can be traced to its source components by decomposing
model predictions to a hierarchy of traceable components. Moreover, the physical emulators
make it computationally possible to assimilate multiple sources of data to constrain predictions
of complex models.
The theoretical framework we developed in this study can well explain dynamics of C
storage in response to cyclic seasonal change in external forcings (e.g., Figs. 2 and 3) and climate
warming and rising atmospheric $CO_2$ (Fig. 5). It also can explain responses of ecosystem C
storage to disturbances and other global change factors, such as nitrogen deposition, land use
changes, and altered precipitation. The theoretical framework is simple and straightforward but
able to characterize the direction and rate of C storage change, which are arguably among the
most critical issues for quantifying terrestrial C sequestration. Future research should explicitly
incorporate stochastic disturbance regime shifts (e.g., Weng et al., 2012) and vegetation
dynamics (Moorcroft et al., 2001; Purves and Pacala, 2008; Fisher et al., 2010; Weng et al.,
2015) into this theoretical framework to explore their theoretical issues related to
biogeochemistry.


**Acknowledgements**: This work was partially done through the working group, Nonautonomous




Systems and Terrestrial Carbon Cycle, at the National Institute for Mathematical and Biological
Synthesis, an institute sponsored by the National Science Foundation, the US Department of
Homeland Security, and the US Department of Agriculture through NSF award no. EF-0832858,
with additional support from the University of Tennessee, Knoxville. Research in Yiqi Luo
EcoLab was financially supported by U.S. Department of Energy grants DE-SC0006982, DE-
SC0008270, DE-SC0014062, DE-SC0004601, and DE-SC0010715 and U.S. National Science
Foundation (NSF) grants DBI 0850290, EPS 0919466, DEB 0840964, and EF 1137293.

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



**Fig. 1** The Terrestrial ECOsystem (TECO) model and its outputs. Panels a is a schematic
representation of C transfers among multiple pools in plant, litter and soil in the TECO model.
TECO has feedback loops of C among soil pools. CWD = coarse wood debris, SOM = Soil
Organic Matter. Panel b compares the original TECO model outputs with those from matrix
equations for net ecosystem production (NEP = the sum of elements in $X'(t)$ from eq. 1).  Panel
c compares the original TECO model outputs with those from matrix equations for ecosystem C
storage (= the sum of elements in $X(t)$ from eq. 2). The C storage values calculated with eq. 2
are close to 1:1 line with $r^2$ =0.998 with the modeled values (panel c). The minor mismatch in
estimated C storage between the matrix equation calculation and TECO outputs is due to
numerical errors via inverse matrix operation with some small numbers.

**Fig. 2** Seasonal cycles of the C storage capacity and C storage dynamics for the leaf pool (i.e.,
pool 1 as shown in Fig. 1). All the components are showed in panels b-d to calculate $x_{c,1}(t) =$
$b_1 u(t) \tau_1$ through multiplication, where $u(t) = NPP$ and $\tau_1 = 1/k_1$ for leaf.

**Fig. 3** Seasonal cycles of the C storage capacity and C storage dynamics for the litter pool (i.e.,
pool 4 as shown in Fig. 1). All the components are showed to calculate
$x_{c,4,u}(t) = \sum_{j=1}^{n} f_{4j} \tau_4 b_j u(t)$ in panels b-e and $x_{c,4,p}(t) = \sum_{j=1,j\neq4}^{n} f_{4j} \tau_4 x'_j(t)$ in panels f-i for
litter. $x_{c,4,u}(t)$ is the maximal amount of C that can transfer from C input to the litter pool.
$x_{c,4,p}(t)$ is the maximal amount of C that can transfer from all the other pools to the litter pool.
This figure is to illustrate the network of pools through which C is distributed.





**Fig. 4** Components of the C storage capacity for litter pool (i.e., pool 4 as shown in Fig. 1).

Component, $x_{c,4,u}(t)$, is the C from C input and component, $x_{c,4,p}(t)$, is the C from all the other

pools to the litter pool. The sum of them is the attractor that determines the direction of C storage

change in pool 4.

**Fig. 5** Transient dynamics of ecosystem C storage in response to climate change in Harvard

Forest. Panel a shows the time courses of the ecosystem C storage capacity, the ecosystem C

storage potential, and ecosystem C storage (i.e., C stock) from 1850 to 2100. Panel b shows time

courses of NPP(t) as C input and ecosystem residence times. Panel c shows correlated changes in

ecosystem C storage potential and net ecosystem production (NEP). Panel d illustrates the

regression between the C storage potential and NEP.

**Fig. 6** The C storage capacity ($x_{c,i}(t)$), the C storage potential ($x_{p,i}(t)$), and C storage ($x_i(t)$) of

individual pools. The potential is nearly zero for those fast turnover pools with short residence

times but very large for those pools with long residence times.

**Fig. 7** The C storage potential of individual pools ($x_{p,i}$) as influenced by net C pool change of

different pools ($x'_i$) in their corresponding rows. The correlation coefficients show the degree of

influences of net C pool change in one pool on the C storage potential of the corresponding pool

through the network of C transfer. Those empty cells indicate no pathways of C transfer between

those pools as indicated in Fig. 1.

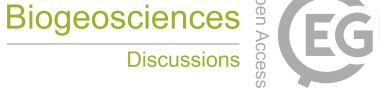



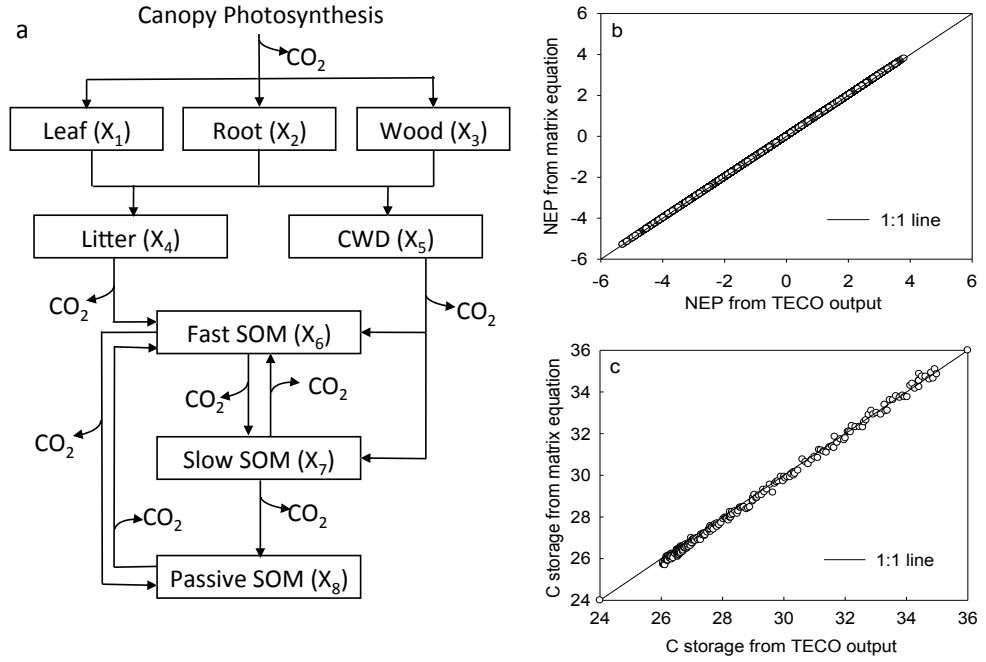

Fig. 1



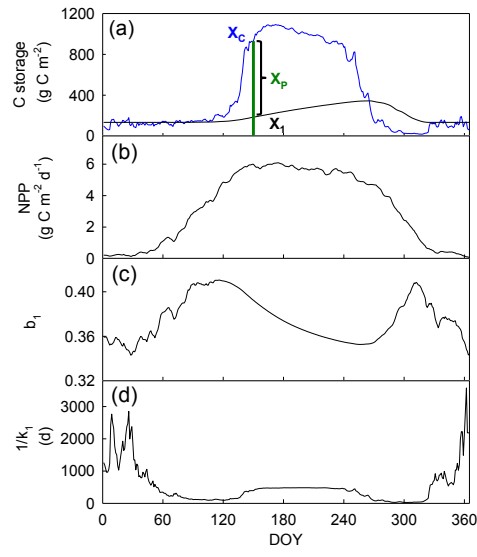

Fig. 2





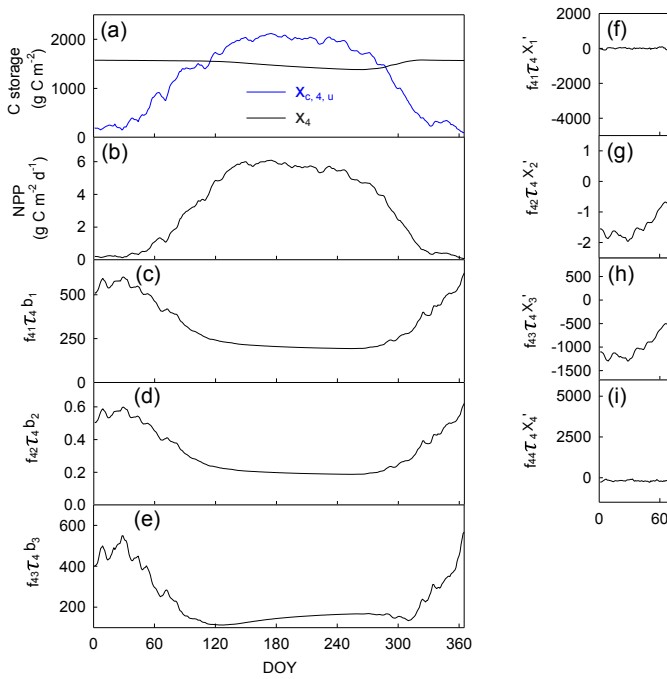

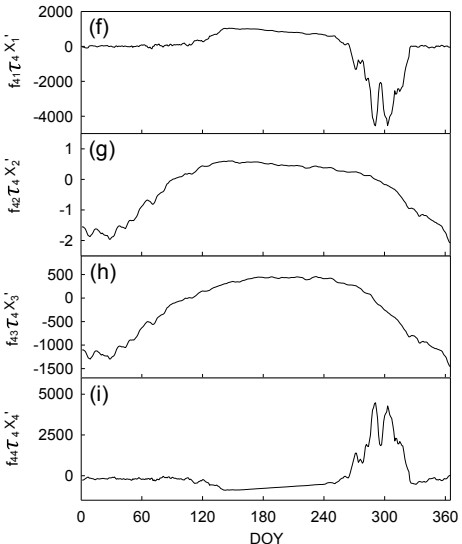

Fig. 3






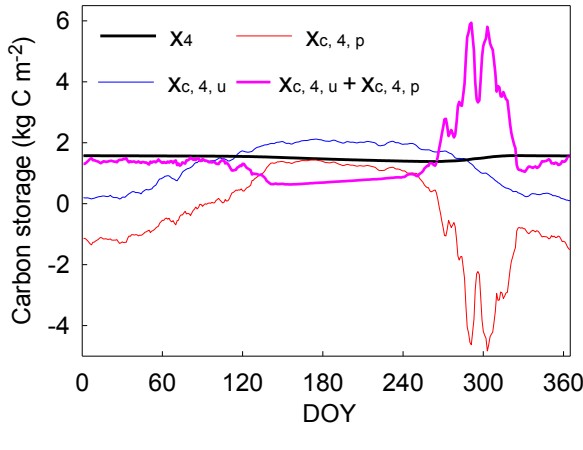

Fig. 4






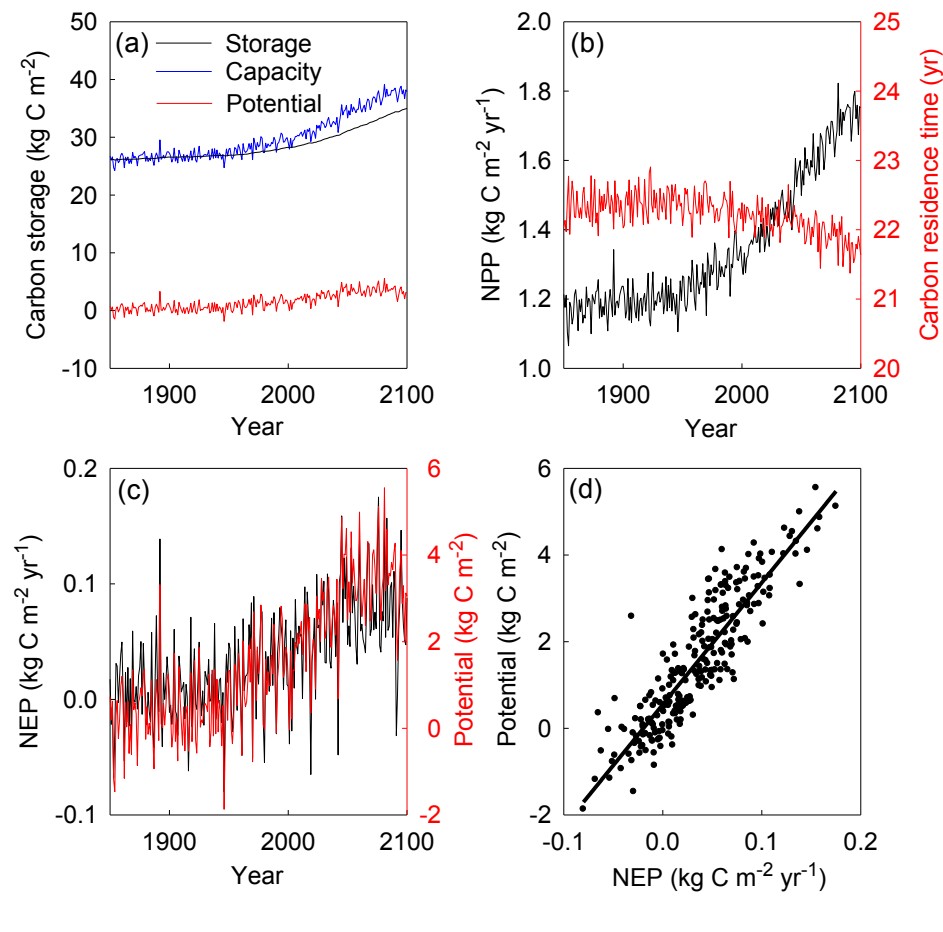

Fig. 5





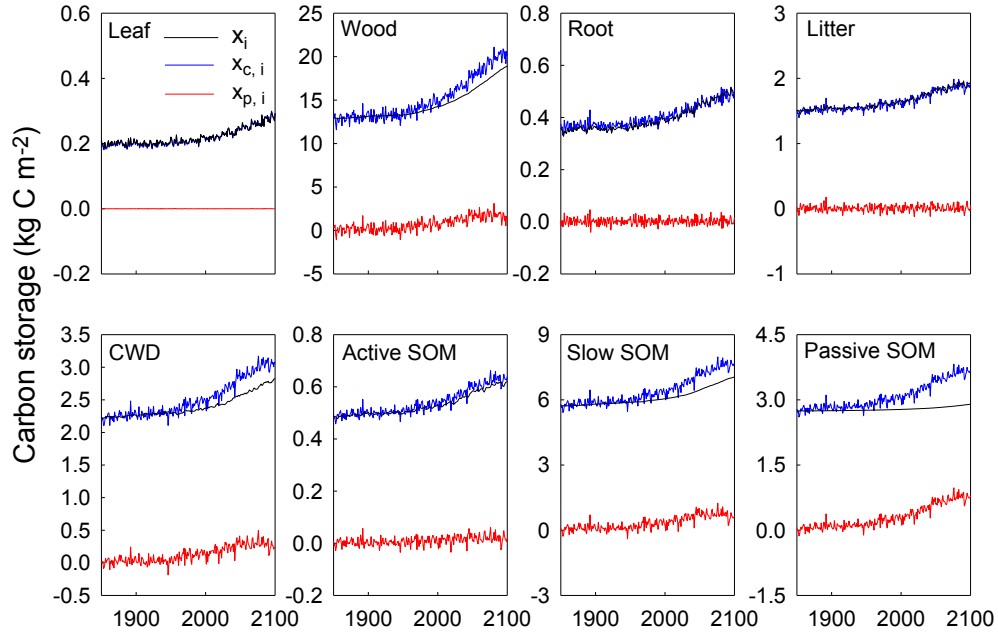

Fig. 6






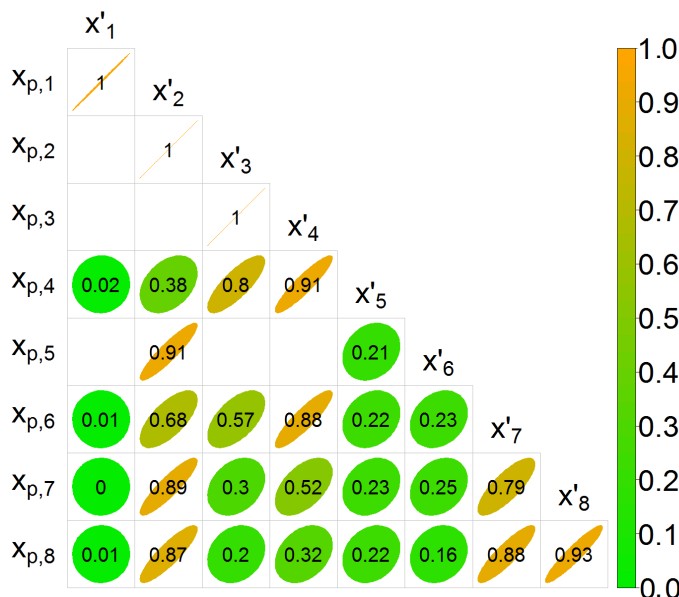


Fig. 7