# Peer review of "Transient Dynamics of Terrestrial Carbon Storage: Mathematical foundation and its applications 3 Yiqi Luo1,2, Zheng Shi1, Xingjie Lu3, Jianyang Xia4, Junyi Liang1, Jiang Jiang1, Ying Wang5, 4 Matthew J. Smith6, Lifen Jiang1, Anders Ahlström7, 8, Benito Chen9, Oleksandra Hararuk10, Alan 5 Hastings11, Forrest Hoffman12, Belinda Medlyn13, Shuli Niu14, Martin Rasmussen15, Katherine 6 Todd-Brown16,"

_Biogeosciences, 2016_

## Referee Comment (RC1) · Anonymous Referee #1 · 27 Oct 2016

General remarks:

The authors present a paper showing that a matrix equation can replicate the output of a comprehensive carbon cycle model. In particular they find that the force driving the ecosystem C storage is the C storage capacity. In general the article is well written and organized and fits into the scope of the journal. Using such a simple matrix equation as a physical emulator of comprehensive models has the potential to save a lot of computing time and gains a deeper understanding of the underlying mechanisms. The authors state in their summary that this would revolutionize model evaluations. I have some concern about this: The matrix equation has to be fitted to a simulation of the complex model with a specified fixed climate scenario. It would be interesting to know

whether this parameter set can be used for a different climate scenario. In particular some parameters in the matrix equation are time-dependent and this time-dependence might change for different climate scenarios. Then the complex model can really be replaced by the matrix equation. Otherwise the matrix equation allows only a more convenient analysis of the model output. Non-linearities in the complex model might lead to a deviation from the linearized matrix representation. It would be nice if the authors could comment on that.

More specific remarks:

Abstract: The authors are talking about a 3-D parameter space. These 3 parameters, however, are not simple scalars, but are itself vectors (e.g., residence time and storage potential).

Page 4: The authors state that most carbon cycle models follow a mathematical formulation of ordinary differential equations. Many of the dynamic global vegetation models (DGVM) are ab initio formulated as a time discrete model calculating, e.g., NPP on a daily level and carbon allocation to different vegetation pools on annually using some (non-linear) allocation rules. Moreover, the authors should mention these DGVMs.

Page 9: The authors should describe which algorithms are necessary in order to develop the matrix equation from the output of the TECO model. In particular how they determined matrix A and K. Technical comment:

Page 29, line 586: A "to" is missing: The emulators allow us TO analyze. . .

In summary the article is suitable for publication if the above-mentioned comments are incorporated.

---

## Short Comment (SC1) · 11 Nov 2016

Luo et al provide an excellent mathematical framework for studying the dynamics of the carbon cycle in terrestrial ecosystems. The focus on transient dynamics makes clear which aspects of carbon storage and sequestration are most important to consider in order to understand the functioning of forests are carbon reservoirs. The reduction of the models to a 3D parameter space is seemingly very useful for a mechanistic understanding of the effects of global change on terrestrial carbon storage.

The modeling assumptions could use further clarification. In particular, the assumption that short-term disturbances can be well represented by the matrix equation (assumption 5) and the assertion that this assumption is unlikely to affect the results need

further support. Disturbances may be very important for the carbon cycling of terres-trial systems and can affect ecosystem dynamics and carbon cycling for decades, in addition to causing C fluxes that greatly exceed those from annual cycles. Presum-ably, disturbance events could be incorporated in the time varying factors u(t) and $\xi(t)$. However, there are a number of well-developed non-linear models for pest outbreaks that might violate the assumption that transfer between pools can be represented by a linear model (assumption 1) if outbreaks were to be incorporated into these factors.

While one aspect of pest outbreaks is a reduction in GPP or NPP, which may be suf-ficiently represented by u(t), even a linear approximation of the rapid change in the transfer of biomass between classes cannot be represented by this model without making the matrix A of transfer coefficients also time-dependent. One way this may be overcome is by setting limits on the timescale of applicability of this mathematical framework, so as to assume that transfer coefficients are not changing. Further, abi-otic disturbances such as fire or disturbances that remove carbon from the ecosystem completely such as harvesting would be outside the scope of this model. The authors show that X'(t) in this model is the net ecosystem production (NEP), but non-biotic transformation from organic and inorganic carbon is not included in NEP, nor is transfer between ecosystems. This may just require a clarification of terminology in order to include fire, other abiotic oxidation, and harvesting in the $\xi(t)$ term of the model.

Finally, it may be useful to clarify on what scale the results apply. Based on the as-sumptions about linear decay smoothing small scale fluctuations and the neglect of lateral C fluxes, it seems important to point out that this is model applies only at the ecosystem scale. The parameters are calibrated based on one grid cell of the TECO model; would the same procedure be expected to scale up to larger spatial scales?

In the conclusion, the authors state that this model is consistent with complex dynamics including tipping points, which they say are "caused by multiple environmental forcing variables interacting with relatively simple internal processes over different temporal and spatial scales." Tipping point behavior crucially depends on non-linear dynamics and so seems inconsistent with this model. However, a clarification that this method

can evaluate the transient dynamics in a given state but does not reproduce more complex behavior may be more accurate.

---

## Referee Comment (RC2) · Anonymous Referee #2 · 16 Nov 2016

Manuscript: **Transient Dynamics of Terrestrial Carbon Storage: Mathematical Foundation and Numeric Examples**
Author(s): **Yiqi Luo et al.**

Evaluation:  rejection

Remarks and comments

In spite of words "mathematical foundation" in the title, the first mistake is contained directly in the first formula (1). Let's rewrite it in the component form:

$$\begin{pmatrix} X_1^{'} \\ ... \\ X_n^{'} \end{pmatrix} = \begin{pmatrix} B_1 \\ ... \\ B_n \end{pmatrix} u(t) - \begin{pmatrix} A_{11} & ... & A_{1n} \\ ... & ... & ... \\ A_{n1} & ... & A_{nn} \end{pmatrix} \begin{pmatrix} \xi_1 & 0 & 0 \\ 0 & ... & 0 \\ 0 & 0 & \xi_n \end{pmatrix} \begin{pmatrix} K_1 & 0 & 0 \\ 0 & ... & 0 \\ 0 & 0 & K_n \end{pmatrix} \begin{pmatrix} X_1 \\ ... \\ X_n \end{pmatrix} \quad (1)$$

and see that in this notation all off-diagonal elements of matrix *A* are useless, and the system (1) is simply a set of trivial linear equations for disconnected variables. Do the authors know that matrix multiplication is non-commutative ? My hypothesis is that the matrix A should be stated after other multipliers in the second member of the sum:

$$X^{'}(t) = Bu(t) - \xi KAX(t).$$

Such a formula is at least mathematically correct and allows the following component view:

$$\begin{pmatrix} X_1^{'} \\ ... \\ X_n^{'} \end{pmatrix} = u(t)\begin{pmatrix} B_1 \\ ... \\ B_n \end{pmatrix} - \begin{pmatrix} \xi_1 & 0 & 0 \\ 0 & ... & 0 \\ 0 & 0 & \xi_n \end{pmatrix} \begin{pmatrix} K_1 & 0 & 0 \\ 0 & ... & 0 \\ 0 & 0 & K_n \end{pmatrix} \begin{pmatrix} A_{11} & ... & A_{1n} \\ ... & ... & ... \\ A_{n1} & ... & A_{nn} \end{pmatrix} \begin{pmatrix} X_1 \\ ... \\ X_n \end{pmatrix} \quad (1\text{-}a)$$

Consequently all next formulas should be corrected according to the new form of (1). It's completely unclear why "all off-diagonal values $a_{ji}$ are negative" (page 8).

But the more essential question is concerned to it's biological correctness and sense. According to (1, 1-a) matrix *A* consists of transfer coefficients and does not depend on system variables *X* making all the system non-autonomous and linear. There is no biological foundation for such strong universality of the form (1, 1-a) for all temporal and spatial scales and no mathematical proof in the paper. In particular, it's not clear how mass-balance relations are connected with that form.

Page 9 gives us an example of a risky statements made in the paper. Authors say that almost all world models of carbon cycle in terrestrial ecosystems have the form (1). They refer to the work (Manzoni, Porporato, 2009) and state that there is a review of 250 models of carbon cycling in it ! First, Table A2 in this work has 200 references to papers describing different versions of a smaller number of models. Second, I have a very strong doubt that all of them can be presented in the form (1) because they were made for various time scales, different set of compartments and different details of biogeochemical processes accounted for. Interesting is the fact that the model of Manzoni and Porporato (2009) themselves is nonlinear and does not look like the system (1) ! As well as another model of soil organic carbon and microbial dynamics made by Hararuk et al. (2015) also referred to by the authors !

In part 2.2 (pages 9-11) authors carry out comparison of the TECO terrestrial ecosystem model results and the system (1) calculations. Their statement on a 100% match of NEE calculations for TECO and (1) seem strange. If TECO is independent of the system (1) this is unbelievable result, in the opposite case the comparison has no sense.

Introducing two new definitions – the C storage capacity and C storage potential – could be a good idea of this paper if authors would explain their biological interpretation and mathematical correctness. First, we should make correspondence to (1-a) and note that $\tau_{ch} = (\xi K A)^{-1}$ instead of (3). Second, study of existence for this inverse matrix is needed to state mathematical correctness of these definitions because inverse matrix serves as a foundation for all math terms in the following text. There is no such study in the paper. Another question arises about chasing time $\tau_{ch}$: why it's formula $\tau_{ch} = (\xi K A)^{-1}$ should have physical dimension of time ? There are no explanations in the text.

All inputs in the model (1) are supposed constant or time-dependent. In particular on page 15 plant photosynthesis is declared only time-dependent. But for some temporal scales (a year, for example) it can essentially dependent on the plant carbon content and in that case the model (1) should have another form (Parolari, Porporato, 2016). Therefore, since all other formulas and descriptions are based on the terms introduced above with mistakes as well as statements made without sufficient biological basis, the conclusion at page 25 (part 4.4, first sentence) about novel approach suggested by the authors to understand, evaluate, diagnose and improve carbon cycle models is represented as inadequate and seems early and premature.

Reference
Parolari A., Porporato A., Forest soil carbon and nitrogen cycles under biomass harvest: stability, transient response, and feedback. // *Ecological Modelling*, v. 329, 2016, pp. 64-76.

Overall conclusion: the manuscript should be rejected.

---

## Author Comment (AC1) · 22 Nov 2016

Dear Referee 1:

We greatly appreciate your comments on our manuscript. We have carefully studied your comments and revised the manuscript accordingly. Please note the line numbers and pages numbers in this letter are all refereed in the revised manuscript.

Hope you will find our revision and responses satisfactory.

Yiqi Luo On behalf of all the authors

Below we list our point-to-point responses to your comments:

[Comment] General remarks: The authors present a paper showing that a matrix equation can replicate the output of a comprehensive carbon cycle model. In particular they find that the force driving the ecosystem C storage is the C storage capacity. In general the article is well written and organized and fits into the scope of the journal. Using such a simple matrix equation as a physical emulator of comprehensive models has the potential to save a lot of computing time and gains a deeper understanding of the underlying mechanisms. The authors state in their summary that this would revolutionize model evaluations.

[Response] Thanks for the positive comment.

[Comment] I have some concern about this: The matrix equation has to be fitted to a simulation of the complex model with a specified fixed climate scenario. It would be interesting to know whether this parameter set can be used for a different climate scenario. In particular some parameters in the matrix equation are time-dependent and this time-dependence might change for different climate scenarios. Then the complex model can really be replaced by the matrix equation. Otherwise the matrix equation allows only a more convenient analysis of the model output. Non-linearities in the complex model might lead to a deviation from the linearized matrix representation. It would be nice if the authors could comment on that.

[Response] The physical emulator does not result from fitting the model to simulation of the complex model. It generates by organizing the carbon balance equations in the original model into a matrix form. So the physical emulator is not climate scenario-specific. Once developed, it is applicable to all climate scenarios.

We have revised the manuscript to clarify this point. For example, we revised the title of section 2.2 to be "TECO Model, its physical emulator, and numerical experiments". We completely rewrote the third paragraph in that section to describe how we have developed the physical emulator of TECO in detail as:

"To support the mathematical analysis using eq. 1, we first developed a physical emu-

lator (i.e., the matrix representation of eq. 1) of the TECO model and then verified that the physical emulator can exactly represent simulations of the original TECO model. We first identified those parameter values in each of the C balance equations in the TECO model that are corresponding to elements in matrices A and K in eq. 1. The time-dependent variables for u(t), elements in vector B, and elements in matrix $\xi(t)$ in the physical emulator were directly from outputs of the original TECO model. Then those parameter values and time-dependent variables were organized into matrices A, $\xi(t)$, and K; vectors X(t), X_0, and B; and variable u(t). Those matrices, vectors, and variable were entered to matrix calculation to compute X'(t) using eq. 1. The sum of elements in calculated X'(t) is a 100% match with simulated net ecosystem production (NEP) with the TECO model (Fig. 1b)."

Hope this paragraph explains the physical emulator clearly. In addition, we will provide a webpage link to both the TECO model and its physical emulator for verification and uses.

[Comment] More specific remarks: Abstract: The authors are talking about a 3-D parameter space. These 3 parameters, however, are not simple scalars, but are itself vectors (e.g., residence time and storage potential).

[Response] we add elements of the vectors together to get the scalars before we plotted the 3D parameter space. We clarified this point in several places in the manuscript. For example,

One paragraph on page 14 (lines 318-322) on this point is:

"Note that sums of elements in vectors X(t), X_c (t), X_p (t), Xˆ' (t), and $\tau$_E (t) are corresponding, respectively, to the whole ecosystem C stock, ecosystem C storage capacity, ecosystem C storage potential, net ecosystem production (NEP), and ecosystem residence time. In this paper, we do not use a separate set of symbols to represent those sums rather than express them wherever necessary. "

Also, the legend of Figure 1 explains this point:

"Panel b compares the original TECO model outputs with those from matrix equations for net ecosystem production (NEP = the sum of elements in X'(t) from eq. 1). Panel c compares the original TECO model outputs with those from matrix equations for ecosystem C storage (= the sum of elements in X(t) from eq. 2)."

[Comment] Page 4: The authors state that most carbon cycle models follow a mathematical formulation of ordinary differential equations. Many of the dynamic global vegetation models (DGVM) are ab initio formulated as a time discrete model calculating, e.g., NPP on a daily level and carbon allocation to different vegetation pools on annually using some (non-linear) allocation rules. Moreover, the authors should mention these DGVMs.

[Response] Thanks for the comments. It is not very clear with "are ab initio formulated." That leaves some uncertainty about our understanding of this comment. Nevertheless, the time steps of NPP calculation and allocation do not affect Eq. 1. Indeed, eq. 1 is mainly about C transformation within land ecosystems before the carbon is respired. NPP is input of eq. 1.

We have successfully applied Eqs. 1 and 2 to LPJ-GUESS, a DGVM, as described in line 613.

[Comment] Page 9: The authors should describe which algorithms are necessary in order to develop the matrix equation from the output of the TECO model. In particular how they determined matrix A and K.

[Response] We wrote the physical emulator of the TECO model in matlab. But it can be developed in any other computer language. Basically, we have to understand the original model and identify those carbon balance equations. Then we organize those coefficients and parameters in matrix forms to develop the physical emulator. See our responses to your comment on emulator above. We have completely revised the

paragraph in Section 2.2 to describe how we developed the physical emulator of the TECO model.

[Comment] Technical comment: Page 29, line 586: A "to" is missing: The emulators allow us TO analyze: : :

[Response] Corrected as suggested.

[Comment] In summary the article is suitable for publication if the above-mentioned comments are incorporated.

[Response] Thank the referee for the support.

―――――――――――――――――――

---

## Author Comment (AC2) · 22 Nov 2016

Dear M. Freilich, maraf@mit.edu:

We greatly appreciate your comments on our manuscript. We have carefully studied your comments and revised the manuscript accordingly. Please note the line numbers and pages numbers in this letter are all refereed in the revised manuscript.

Hope you will find our revision and responses satisfactory.

Yiqi Luo On behalf of all the authors

Below we list our point-to-point responses to your comments:

[Figure]

[Comment] Luo et al provide an excellent mathematical framework for studying the dynamics of the carbon cycle in terrestrial ecosystems. The focus on transient dynamics makes clear which aspects of carbon storage and sequestration are most important to consider in order to understand the functioning of forests are carbon reservoirs. The reduction of the models to a 3D parameter space is seemingly very useful for a mechanistic understanding of the effects of global change on terrestrial carbon storage.

[Response] We greatly appreciate your positive comments.

[Comment] The modeling assumptions could use further clarification. In particular, the assumption that short-term disturbances can be well represented by the matrix equation (assumption 5) and the assertion that this assumption is unlikely to affect the results need further support. Disturbances may be very important for the carbon cycling of terrestrial systems and can affect ecosystem dynamics and carbon cycling for decades, in addition to causing C fluxes that greatly exceed those from annual cycles.

[Response] We agree. Disturbances can substantially affect ecosystem carbon cycling

[Comment] Presumably, disturbance events could be incorporated in the time varying factors u(t) and _(t). However, there are a number of well-developed non-linear models for pest outbreaks that might violate the assumption that transfer between pools can be represented by a linear model (assumption 1) if outbreaks were to be incorporated into these factors.

[Response] We appreciate for your point that there are many non-linear models for pest outbreaks. Pest outbreaks affect tree mortality, which usually is in proportion to the severity of pest outbreaks. Tree mortality can be non-linearly responding to pest outbreaks as decomposition of soil organic carbon to temperature. Such non-linear responses still do not affect fundamental properties of the carbon cycle as discussed in Assumption 4 on response functions.

[Comment] While one aspect of pest outbreaks is a reduction in GPP or NPP, which may be sufficiently represented by u(t), even a linear approximation of the rapid change in the transfer of biomass between classes cannot be represented by this model without making the matrix A of transfer coefficients also time-dependent. One way this may be overcome is by setting limits on the timescale of applicability of this mathematical framework, so as to assume that transfer coefficients are not changing. Further, abiotic disturbances such as fire or disturbances that remove carbon from the ecosystem completely such as harvesting would be outside the scope of this model.

[Response] Matrix A can be time-dependent. Equation 1 does not explicitly include abiotic disturbances in influencing carbon cycle. Weng et al. (2012) developed a disturbance regime model that explicitly incorporates disturbances into equation 1 for their influences of terrestrial carbon cycle. This paper focuses on understanding of fundamental properties of equation 1.

To clarify this point, we have revised the second half of the paragraph on Assumption 4 as:

"Those disturbance influences can be represented in terrestrial C cycle models through changes in parameter values, environmental scalars, and/or discrete C transfers among pools of eq. 1 (Luo and Weng 2011). While eq. 1 does not explicitly incorporate disturbances for their influences on land C cycle, Weng et al. (2012) developed a disturbance regime model that combines eq. 1 with frequency distributions of disturnace severity and intervals to quantify net biome exchanges."

[Comment] The authors show that X'(t) in this model is the net ecosystem production (NEP), but non-biotic transformation from organic and inorganic carbon is not included in NEP, nor is transfer between ecosystems. This may just require a clarification of terminology in order to include fire, other abiotic oxidation, and harvesting in the _(t) term of the model.

[Response] Yes, you are very sharp to point out the omission of this analysis. We did

not explicitly include disturbances in the analysis but state that disturbances do not alter fundamental properties of the system. As explained above, Weng et al. (2012) developed a model that explicitly combines disturbances with equation 1 to quantify net biome production on lines 503-506.

[Comment] Finally, it may be useful to clarify on what scale the results apply. Based on the assumptions about linear decay smoothing small scale fluctuations and the neglect of lateral C fluxes, it seems important to point out that this is model applies only at the ecosystem scale. The parameters are calibrated based on one grid cell of the TECO model; would the same procedure be expected to scale up to larger spatial scales?

[Response] Thanks for your comment. Equation 1 has been also applied to several global models, such as National Center for Atmosphere Research (NCAR) Community Land Model (CLM) and LPJ-GUESS. See a published paper by Ahlström et al. (2015) for the application of equation to the global model LPJ-GUESS. Fundamentally equation 1 fully represents carbon balance equations in matrix form for almost all the land carbon cycle models. Equation 1 does not do any more smoothing of small-scale fluctuations than do the original models. The paragraph on page 27 about physical emulators explains it.

Yes, equation 1 does not apply to the models with lateral fluxes.

[Comment] In the conclusion, the authors state that this model is consistent with complex dynamics including tipping points, which they say are "caused by multiple environmental forcing variables interacting with relatively simple internal processes over different temporal and spatial scales." Tipping point behavior crucially depends on non-linear dynamics and so seems inconsistent with this model. However, a clarification that this method can evaluate the transient dynamics in a given state but does not reproduce more complex behavior may be more accurate.

[Response] You are right that the eq. 1 does not cause some of the complex dynamics such as tipping points. Tipping points occur in carbon cycle mainly due to complex behaviors in external forcings. Luo and Weng (2011) and Luo et al. (2015) have explained this phenomenon in detail. While this paper could not explain this in detail again, we revised the manuscript by pointing readers to those papers for detailed discussion as on pages 29-30:

"The two components of land C storage dynamics represent interactions of external forces (via changes in the capacity) and internal capability of the land C cycle (via changes in the C storage potential) to generate complex phenomena of C cycle dynamics, such as fluctuations, directional changes, and tipping points, in the terrestrial ecosystems. From a system perspective, these complex phenomena could not be generated by relatively simple internal processes but are mostly caused by multiple environmental forcing variables interacting with internal processes over different temporal and spatial scales as explained by Luo and Weng (2011) and Luo et al. (2015). Note that while those internal processes can be mathematically represented with a relatively simple formula, their ecological and biological underpinnings can be very complex."

---

## Author Comment (AC3) · 22 Nov 2016

Dear Referee 2:

We greatly appreciate your time and effort to read, understand, and make comments on our manuscript. We have carefully studied your comments and revised the manuscript accordingly. Hope our responses have adequately addressed your concerns so that we can develop mutual understanding about your concerns and about what we present in the paper.

Please note the line numbers and pages numbers in this letter are all refereed in the revised manuscript.

Yiqi Luo
On behalf of all the authors

Below we list our point-to-point responses to your (i.e., referee 2 in this case) comments:

[Comment] In spite of words "mathematical foundation" in the title, the first mistake is contained directly in the first formula (1). Let's rewrite it in the component form:

$$\begin{pmatrix} X_1' \\ ... \\ X_n' \end{pmatrix} = \begin{pmatrix} B_1 \\ ... \\ B_n \end{pmatrix} u(t) - \begin{pmatrix} A_{11} & ... & A_{1n} \\ ... & ... & ... \\ A_{n1} & ... & A_{nn} \end{pmatrix} \begin{pmatrix} \xi_1 & 0 & 0 \\ 0 & ... & 0 \\ 0 & 0 & \xi_n \end{pmatrix} \begin{pmatrix} K_1 & 0 & 0 \\ 0 & ... & 0 \\ 0 & 0 & K_n \end{pmatrix} \begin{pmatrix} X_1 \\ ... \\ X_n \end{pmatrix} \tag{1}$$

and see that in this notation all off-diagonal elements of matrix $A$ are useless, and the system (1) is simply a set of trivial linear equations for disconnected variables. Do the authors know that matrix multiplication is non-commutative? My hypothesis is that the matrix A should be stated after other multipliers in the second member of the sum:

$$X'(t) = Bu(t) - \xi K A X(t)$$

Such a formula is at least mathematically correct and allows the following component view:

$$\begin{pmatrix} X_1' \\ ... \\ X_n' \end{pmatrix} = u(t) \begin{pmatrix} B_1 \\ ... \\ B_n \end{pmatrix} - \begin{pmatrix} \xi_1 & 0 & 0 \\ 0 & ... & 0 \\ 0 & 0 & \xi_n \end{pmatrix} \begin{pmatrix} K_1 & 0 & 0 \\ 0 & ... & 0 \\ 0 & 0 & K_n \end{pmatrix} \begin{pmatrix} A_{11} & ... & A_{1n} \\ ... & ... & ... \\ A_{n1} & ... & A_{nn} \end{pmatrix} \begin{pmatrix} X_1 \\ ... \\ X_n \end{pmatrix} \tag{1-a}$$

Consequently all next formulas should be corrected according to the new form of (1). It's completely unclear why "all off-diagonal values $a_{ji}$ are negative" (page 8).

[Response] We are grateful to you for your time and effort to examine mathematical formulas. We agree with you that it is critical to make sure the mathematical expression of biological processes should be correct before we do any analysis.

Your comment prompted us to carefully re-examine the equation. After the multiplication of $\xi$,

$K$ and $A$, Equation 1 becomes:

$$\begin{bmatrix} X_1' \\ X_2' \\ \cdots \\ X_n' \end{bmatrix} = \begin{bmatrix} B_1 \\ B_2 \\ \cdots \\ B_n \end{bmatrix} u(t) - \begin{bmatrix} A_{11}\xi_1 K_1 & A_{12}\xi_2 K_2 & \cdots & A_{1n}\xi_n K_n \\ A_{21}\xi_1 K_1 & A_{22}\xi_2 K_2 & \cdots & A_{2n}\xi_n K_n \\ \cdots & \cdots\cdots\cdots & \cdots & \cdots \\ A_{n1}\xi_1 K_1 & A_{n2}\xi_2 K_2 & \cdots & A_{nn}\xi_n K_n \end{bmatrix} \begin{bmatrix} X_1 \\ X_2 \\ \cdots \\ X_n \end{bmatrix}$$

Then the carbon dynamics in pool 1 will be described by:

$$X_1' = B_1 u(t) - (A_{11}\xi_1 K_1 X_1 + A_{12}\xi_2 K_2 X_2 + \cdots + A_{1n}\xi_n K_n X_n)$$

The above equation states that change in carbon content in pool 1 equals carbon influx from a fraction of NPP (i.e., $u(t)$ times partitioning coefficient $B_1$) minus decomposition expressed by $(A_{11}\xi_1 K_1 X_1 + A_{12}\xi_2 K_2 X_2 + \cdots + A_{1n}\xi_n K_n X_n)$. Since K is decomposition coefficient, the term $K_1 X_1$ describes that decomposition of carbon in pool 1 equals $K_1$ times $X_1$, so on for $K_2 X_2$, and $K_n X_n$. Environmental scalar $\xi_i$ modifies its corresponding $K_i$. Transfer coefficient $A_{1j}$ in the above equation describes carbon transfer from pool j to pool 1. In the real world, no carbon is transferred from other plant, litter, and soil pools to leaf pool. Thus $A_{1j} = 0, j \neq 1$.

However, not all $A_{ij} = 0, j \neq i$. In TECO model with carbon transfer pathways as depicted in Figure 1a, $A_{41} \neq 0$ as it represents litterfall from leaf pool to metabolic litter pool. Xia et al. (2012) explicitly described the A matrix with all elements for CABLE model. There are many zero but several non-zero elements in matrix $A$ to represent carbon transfers among pools. Many of those none-zero transfer coefficients as represented by $A_{ij}$ are related to microbial carbon use efficiency.

Let us look at the equation you suggested (i.e., Equation 1-a). After the multiplication of $\xi$, $K$ and $A$, Equation 1-a will become:

$$\begin{bmatrix} X_1' \\ X_2' \\ \cdots \\ X_n' \end{bmatrix} = u(t) \begin{bmatrix} B_1 \\ B_2 \\ \cdots \\ B_n \end{bmatrix} - \begin{bmatrix} A_{11}\xi_1 K_1 & A_{12}\xi_1 K_1 & \cdots & A_{1n}\xi_1 K_1 \\ A_{21}\xi_2 K_2 & A_{22}\xi_2 K_2 & \cdots & A_{2n}\xi_2 K_2 \\ \cdots & \cdots\cdots\cdots & \cdots & \cdots \\ A_{n1}\xi_n K_n & A_{n2}\xi_n K_n & \cdots & A_{nn}\xi_n K_n \end{bmatrix} \begin{bmatrix} X_1 \\ X_2 \\ \cdots \\ X_n \end{bmatrix}$$

and the carbon dynamics in pool 1 will be described by:

$$X_1' = B_1 u(t) - (A_{11}\xi_1 K_1 X_1 + A_{12}\xi_1 K_1 X_2 + \cdots + A_{1n}\xi_1 K_1 X_n)$$

In the above equation, the term $K_1 X_2$ or $K_1 X_n$ biologically does not make sense as it describes the decomposition of carbon in pool 2 by coefficient $K_1$. The latter describes the relative rate of decomposition of pool 1. Nor does $K_1 X_n$ biologically make sense.

Thus, we hope that you can see that our original equation still works.

The statement "all off-diagonal values $a_{ji}$ are negative" (page 8) because carbon transfer from

pool i to pool j to be positive by having negative coefficient multiplied with negative sign for this term. We have clarified this point by revising the sentences on line 177-179 as:

"In eq. 1, all the off-diagonal $a_{ji}$ values are negative to neutralize the minus sign to indicate positive C influx to the receiving pools"

[Comment] But the more essential question is concerned to it's biological correctness and sense. According to (1, 1-a) matrix *A* consists of transfer coefficients and does not depend on system variables *X* making all the system non-autonomous and linear. There is no biological foundation for such strong universality of the form (1, 1-a) for all temporal and spatial scales and no mathematical proof in the paper. In particular, it's not clear how mass-balance relations are connected with that form.

[Response] Thanks for your question about the biological basis of the mathematical equation. The two paragraphs from line 144 to 162 describe the biological basis as below:

"Hundreds of models have been developed to simulate terrestrial C cycle (Manzoni and Porporato, 2009). All the models have to simulate processes of photosynthetic C input, C allocation and transformation, and respiratory C loss. It is well understood that photosynthesis is a primary pathway of C flow into land ecosystems. Photosynthetic C input is usually simulated according to carboxylation and electron transport rates (Farquhar et al., 1980). Ecosystem C influx varies with time and space mainly due to variations in leaf photosynthetic capacity, leaf area index of canopy, and a suite of environmental factors such as temperature, radiation, and relative humidity (or other water-related variables) (Potter et al., 1993; Sellers et al., 1996; Keenan et al., 2012; Walker et al., 2014).

Photosynthetically assimilated C is partly used for plant biomass growth and partly released back into the atmosphere through plant respiration. Plant biomass in leaves and fine roots usually lives for several months up to a few years before death, while woody tissues may persist for hundreds of years in forests. Dead plant materials are transferred to litter pools and decomposed by microorganisms to be partially released through heterotrophic respiration and partially stabilized to form soil organic matter (SOM). SOM can store C in the soil for hundreds or thousands of years before it is broken down to $CO_2$ through microbial respiration (Luo and Zhou, 2006). This series of C cycle processes has been represented in most ecosystem models with multiple pools linked by C transfers among them (Jenkinson et al., 1987; Parton et al., 1987; 1988; 1993), including those embedded in earth system models (Ciais et al., 2013). "

Moreover, we have conducted many synthesis studies to examine different aspects of the biological basis. The carbon input via canopy photosynthesis as described by $\begin{bmatrix} B_1 \\ B_2 \\ ... \\ B_n \end{bmatrix} u(t)$ has been well accepted. Scientists in the community have questioned carbon transformation through $A\xi KX$ in equation 1. We examine six assumptions of those carbon cycle models and the validity of our analysis in section 4.1 on pages 19-23. We would be happy to answer any specific questions you would have regarding those assumptions.

[Comment] Page 9 gives us an example of a risky statements made in the paper. Authors say that

almost all world models of carbon cycle in terrestrial ecosystems have the form (1). They refer to the work (Manzoni, Porporato, 2009) and state that there is a review of 250 models of carbon cycling in it ! First, Table A2 in this work has 200 references to papers describing different versions of a smaller number of models. Second, I have a very strong doubt that all of them can be presented in the form (1) because they were made for various time scales, different set of compartments and different details of biogeochemical processes accounted for. Interesting is the fact that the model of Manzoni and Porporato (2009) themselves is nonlinear and does not look like the system (1) ! As well as another model of soil organic carbon and microbial dynamics made by Hararuk et al. (2015) also referred to by the authors !

[Response] Thanks for your comment. We agree with you that the nonlinear microbial models by Manzoni and Porporato (2009) or Hararuk et al. (2015) could not be represented by equation 1. This issue is pointed out in section 4.1 regarding those microbial models (i.e., assumption 1). We also pointed out that thousands of datasets we have reviewed do not seem to support those nonlinear microbial models as described on pages 19-20. Paper by Sierra and Müller (2015) also stated that most of the land carbon cycle models can be represented by equation 1.

Indeed, we have worked with many modeling groups and organized those carbon balance equations in their models into the matrix equations. It has been demonstraed that the matrix equation can represent those original models well as described in paragraph from lines 612-621. Please see another publication by Ahlström et al. (2015) with LPJ-GUESS for the application of eq. 1.

[Comment] In part 2.2 (pages 9-11) authors carry out comparison of the TECO terrestrial ecosystem model results and the system (1) calculations. Their statement on a 100% match of NEE calculations for TECO and (1) seem strange. If TECO is independent of the system (1) this is unbelievable result, in the opposite case the comparison has no sense.

[Response] This is the case. We organized those carbon balance equations of TECO into the matrix equation. We run the matrix equation to get the exact simulation outputs as from the original TECO model. We have done that with CABLE (Xia et al. 2012, 2013). CLM3.5 (Rafiqee et al. 2016), CLM4.5 (Shi et al. in prep.), BEPS (Chen et al. 2016), and LPJ-GUESS (Ahlström et al. 2015). In all the cases, the matrix equation can 100% reproduce simulations of those original models. It is unbelieveable. We understand it is surprising.

[Comment] Introducing two new definitions – the C storage capacity and C storage potential – could be a good idea of this paper if authors would explain their biological interpretation and mathematical correctness. First, we should make correspondence to (1-a) and note that $\tau_{ch} = (\xi KA)^{-1}$ instead of (3). Second, study of existence for this inverse matrix is needed to state mathematical correctness of these definitions because inverse matrix serves as a foundation for all math terms in the following text. There is no such study in the paper. Another question arises about chasing time $\tau_{ch}$: why it's formula $\tau_{ch} = (\xi KA)^{-1}$ should have physical dimension of time ? There are no explanations in the text.

[Response] The biological interpretation of C storage capacity is given in Abstract (Line 42-45, 46-47), Results (lines 259-270), Discussion section 4.2, and Conclusions. For example, sentences

on lines 632-635 in the Conclusion section state:

"The capacity, which is the product of C input and residence time, represents their instantaneous responses to a state of external forcing at a given time. Thus, the C storage capacity quantifies the maximum amount of C that an ecosystem can store at the given environmental condition at a point of time."

Similarly, C storage potential is also biologically explained in Abstract, Results, Discussion, and Conclusions sections. For example, the first paragraph in section 4.3 is:

"The C storage potential represents the internal capability to equilibrate the current C storage with the capacity. Bogeochemically, the C storage potential represents re-distribution of net C pool change, $X'(t)$, of individual pools through a network of pools with different residence times as connected by C transfers from one pool to the others through all the pathways. The potential is conceptually equivalent to the magnitude of disequilibrium as discussed by Luo and Weng (2011)."

Thanks for your comment. We have added the time dimension for chasing time on lines 258-259 as:

"In eq. 2, we name the term $(A\xi(t)K)^{-1}$ the chasing time, $\tau_{ch}(t)$, with a time unit used in exit rate $K$."

[Comment] All inputs in the model (1) are supposed constant or time-dependent. In particular on page 15 plant photosynthesis is declared only time-dependent. But for some temporal scales (a year, for example) it can essentially dependent on the plant carbon content and in that case the model (1) should have another form (Parolari, Porporato, 2016).

Reference
Parolari A., Porporato A., Forest soil carbon and nitrogen cycles under biomass harvest: stability, transient response, and feedback. // *Ecological Modelling*, v. 329, 2016, pp. 64-76.

[Response] We have carefully studied the paper by Parolari and Porporato (2016), particularly that paragraph on NPP on page 66. That study differentiated the productivity regime into C-limited and N-limited. The C-limit regime accounts for limitation of light, temperature and moisture whereas the N-limited regime accounts for nitrogen limitation. Both of the regimes have been discussed in relation with eq. 1. Please see sentences on lines 148-152 and lines 490-493 for more explanation. In the revised manuscript, we cited the paper and explained those environmental factors as represented by scalars on line 152.

[Comment] Therefore, since all other formulas and descriptions are based on the terms introduced above with mistakes as well as statements made without sufficient biological basis, the conclusion at page 25 (part 4.4, first sentence) about novel approach suggested by the authors to understand, evaluate, diagnose and improve carbon cycle models is represented as inadequate and seems early and premature.

[Response] We hope our responses to your comments above can help us communicate well with

you and then gain mutual understanding on what we presented in this paper and what you were concerned.

We thank you for the valuable comments, some of which led us to improve the text and better communicate our points to the reader. We hope our responses above also demonstrate that our formulation did not include mistakes and that the terms we introduced were founded on sound biological principles. Thus, we stand by our conclusion that the presented approach enables one to understand, evaluate, diagnose and improve carbon cycles models.